# Platelets reduce anoikis and promote metastasis by activating YAP1 signaling

Monika Haemmerle[1,2], Morgan L. Taylor[1], Tony Gutschner[3,4], Sunila Pradeep[1], Min Soon Cho[5], Jianting Sheng[6], Yasmin M. Lyons[1], Archana S. Nagaraja[1], Robert L. Dood[1], Yunfei Wen[1], Lingegowda S. Mangala[1,7], Jean M. Hansen[1], Rajesha Rupaimoole[1], Kshipra M. Gharpure[1], Cristian Rodriguez-Aguayo [7,8], Sun Young Yim[9], Ju-Seog Lee[9], Cristina Ivan [8], Wei Hu[1], Gabriel Lopez-Berestein[7,8], Stephen T. Wong[6], Beth Y. Karlan[10], Douglas A. Levine [11], Jinsong Liu[12], Vahid Afshar-Kharghan[5] & Anil K. Sood[1,7,13]

Thrombocytosis is present in more than 30% of patients with solid malignancies and correlates with worsened patient survival. Tumor cell interaction with various cellular components of the tumor microenvironment including platelets is crucial for tumor growth and metastasis. Although it is known that platelets can infiltrate into tumor tissue, secrete pro-angiogenic and pro-tumorigenic factors and thereby increase tumor growth, the precise molecular interactions between platelets and metastatic cancer cells are not well understood. Here we demonstrate that platelets induce resistance to anoikis in vitro and are critical for metastasis in vivo. We further show that platelets activate RhoA-MYPT1-PP1-mediated YAP1 dephosphorylation and promote its nuclear translocation which induces a pro-survival gene expression signature and inhibits apoptosis. Reduction of *YAP1* in cancer cells in vivo protects against thrombocytosis-induced increase in metastasis. Collectively, our results indicate that cancer cells depend on platelets to avoid anoikis and succeed in the metastatic process.

[1] Department of Gynecologic Oncology and Reproductive Medicine, The University of Texas MD Anderson Cancer Center, Houston, TX 77030, USA. [2] Institute of Pathology, Martin-Luther-University Halle-Wittenberg, Halle (Saale), Saxony-Anhalt 06112, Germany. [3] Department of Genomic Medicine, The University of Texas MD Anderson Cancer Center, Houston, TX 77054, USA. [4] Faculty of Medicine, Martin-Luther-University Halle-Wittenberg, Halle (Saale), Saxony-Anhalt 06120, Germany. [5] Section of Benign Hematology, The University of Texas MD Anderson Cancer Center, Houston, TX 77030, USA. [6] Department of Systems Medicine and Bioengineering, Houston Methodist Research Institute, Houston Methodist Hospital, Houston, TX 77030, USA. [7] Center for RNA Interference and Non-Coding RNA, The University of Texas MD Anderson Cancer Center, Houston, TX 77054, USA. [8] Department of Experimental Therapeutics, The University of Texas MD Anderson Cancer Center, Houston, TX 77054, USA. [9] Department of Systems Biology, The University of Texas MD Anderson Cancer Center, Houston, TX 77054, USA. [10] Department of Obstetrics and Gynecology, Cedars-Sinai Medical Center, Geffen School of Medicine at UCLA, Los Angeles, CA 90048, USA. [11] Department of Gynecologic Oncology, Laura and Isaac Perlmutter Cancer Centre, NYU Langone Medical Center, New York, NY 10016, USA. [12] Department of Pathology, The University of Texas MD Anderson Cancer Center, Houston, TX 77030, USA. [13] Department of Cancer Biology, The University of Texas MD Anderson Cancer Center, Houston, TX 77054, USA. Correspondence and requests for materials should be addressed to V.A-K. (email: vakharghan@mdanderson.org) or to A.K.S. (email: asood@mdanderson.org)

**M**etastases are the major cause of death in cancer patients. In ovarian cancer, survival of patients is substantially worse in the presence of metastatic disease and has not been significantly improved over the last ten years[1]. Although it is believed that ovarian cancer mainly metastasizes via the peritoneal cavity, we recently discovered that ovarian cancer cells also spread hematogenously with a high predilection for the omentum[2]. Key regulatory signals for metastasis originate from the interaction between cancer cells and the cellular elements of the tumor microenvironment, including cancer-associated fibroblasts, immune cells and endothelial cells. Additionally, tumor cells interact with platelets both inside the tumor microenvironment and in the blood stream or ascites[3]. Moreover, we recently demonstrated that ovarian cancer cells can activate platelets by secreting ADP[4], thereby stimulating the release of a plethora of growth factors and cytokines[5] and promote tumor

growth. In fact, thrombocytosis (platelet counts > 450,000/ml according to NHLBI) is predictive of poor survival in ovarian[3], pancreatic[6], gastrointestinal[7], breast[8] and lung[9] cancers. The paraneoplastic thrombocytosis results from a paracrine circuit of thrombopoietic cytokines induced by ovarian cancer in the host[3]. Thus, it is suggested that the communication between platelets and cancer cells in the tumor microenvironment, blood stream, and peritoneal fluid has an important role in tumor progression and metastasis. Moreover, elucidation of mechanisms involved in platelet-enhanced metastasis could lead to new approaches for disrupting platelet-dependent tumor cell survival without affecting physiological platelet functions. In the current study, we demonstrate that platelet-cancer cell interaction is important in cancer cells' ability to overcome detachment-induced apoptosis (known as anoikis), which is a major hallmark of metastasis[10]. Our experimental findings implicate a crucial role for platelets in

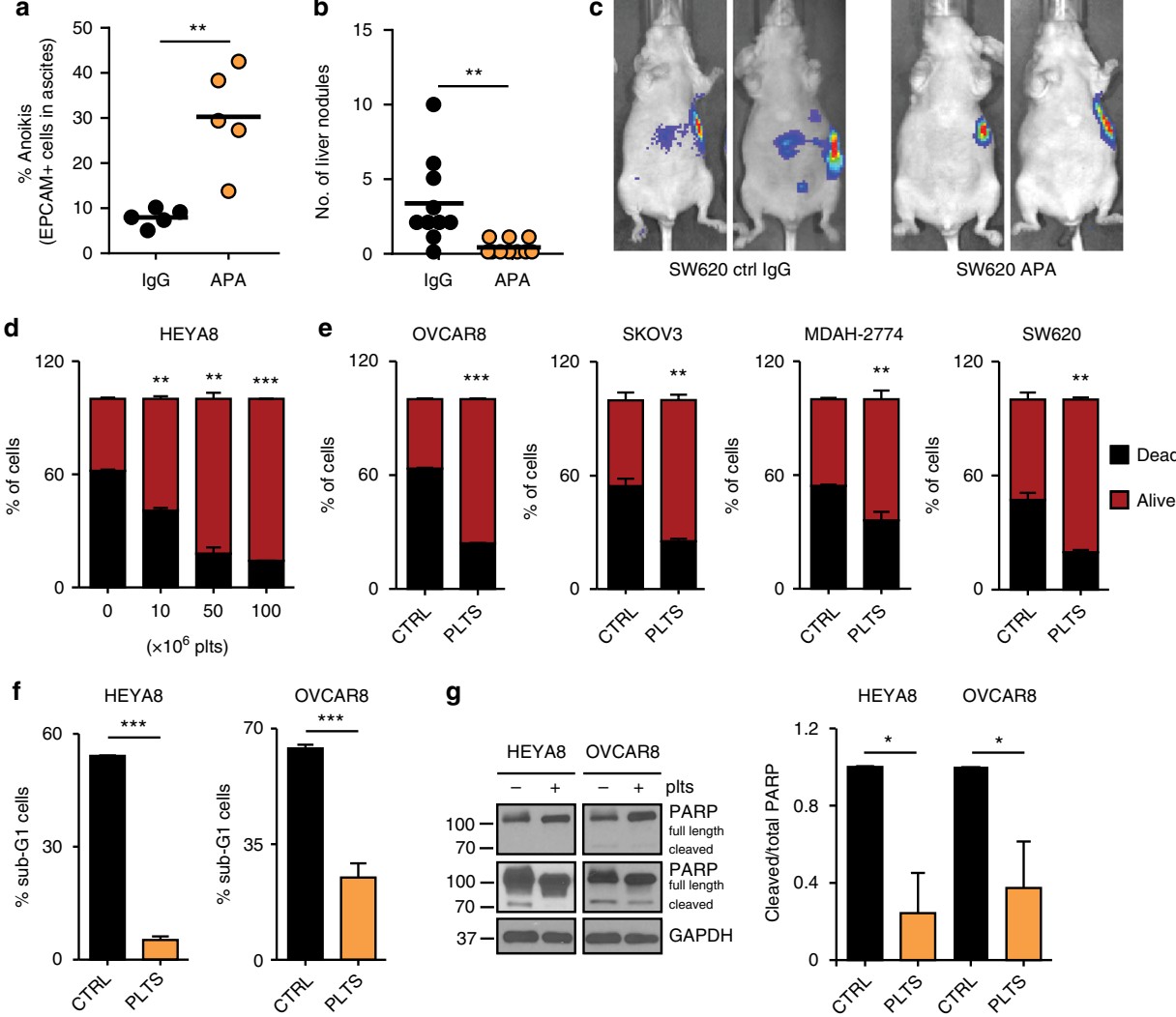

**Fig. 1** Platelets facilitate metastasis in vivo and reduce anoikis. **a** Plot showing percent of SYTOX Red positive EPCAM + MDAH-2774 tumor cells in ascites after treatment with control IgG or anti-platelet antibody (APA, $n = 5$, two-sided Student's $t$-test). **b**, **c** Number of liver nodules **b** and representative bioluminescence imaging pictures **c** 5 weeks after intrasplenic injection of $2 \times 10^6$ SW620 colon cancer cells ($n = 10$, two-sided Student's $t$-test). **d** Number of dead (SYTOX Red positive, *black*) and living (SYTOX Red negative, *red*) HEYA8 cells after 72 hours of low attachment and/or co-incubation with increasing numbers of platelets ($n = 3$, two-sided Student's $t$-test). **e** Number of dead (SYTOX\ Red positive, *black*) and living (SYTOX Red negative, *red*) OVCAR8, SKOV3, MDAH-2774 and SW620 cells after 72 h of low attachment and/or co-incubation with $100 \times 10^6$ platelets ($n = 3$, two-sided Student's $t$-test). **f** Percentage of HEYA8 or OVCAR8 human ovarian cancer cells in sub-G1 phase of the cell cycle after 72 h under low-attachment conditions ($n = 3$, two-sided Student's $t$-test). **g** Protein analysis and quantification of full length and cleaved PARP in HEYA8 and OVCAR8 cells after 72 h under low-attachment conditions (short exposure of PARP *upper panel*, long exposure of PARP *lower panel*, $n = 3$, two-sided Student's $t$-test). GAPDH was used as a loading control. *Bars* and *error bars* represent mean values and the corresponding SEMs ($*p < 0.05$, $**p < 0.01$, $***p < 0.001$)

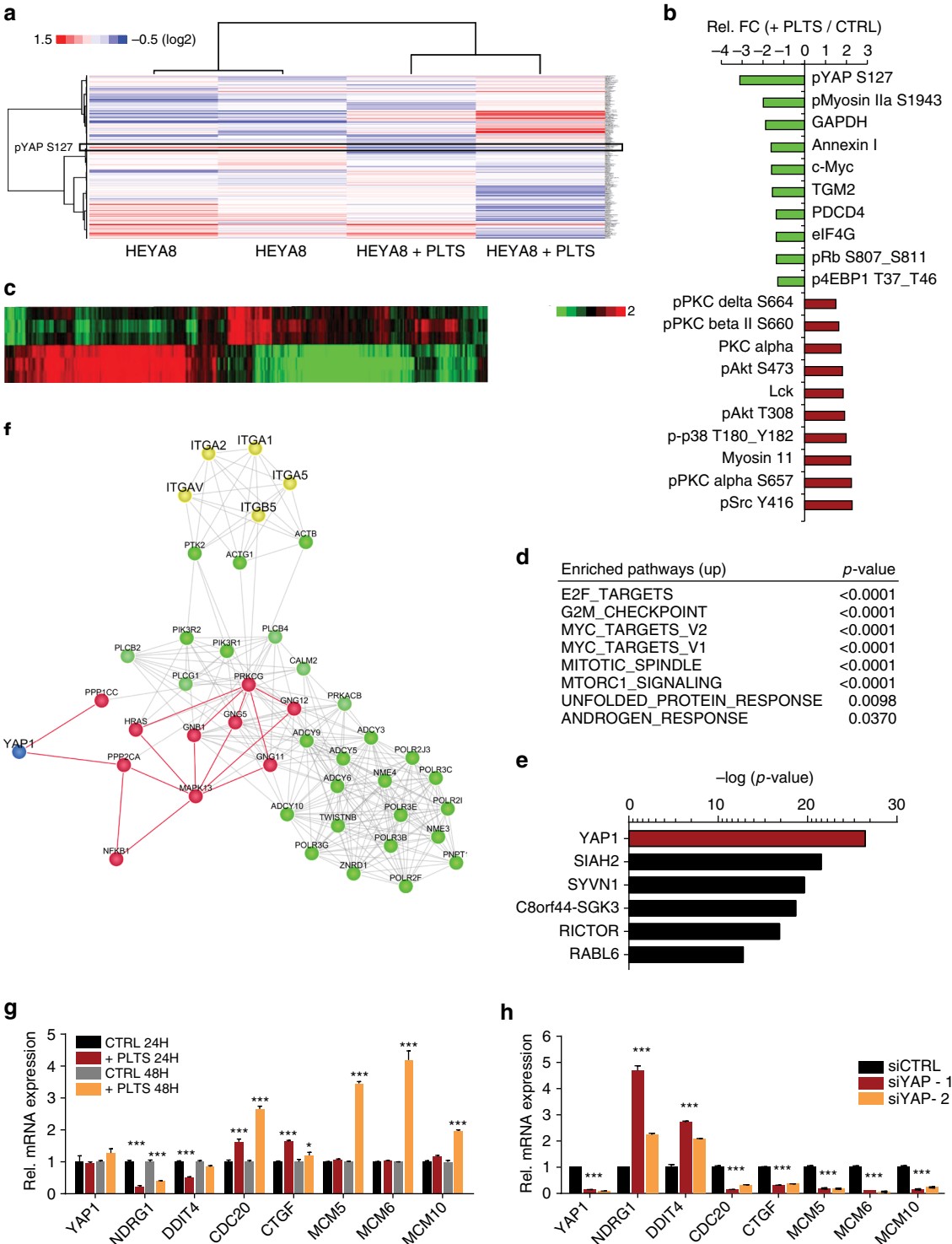

**Fig. 2** Platelets induce a YAP1-specific gene signature in cancer cells. **a** Heat map showing differentially regulated proteins as analyzed by reverse phase protein array (RPPA) in HEYA8 and HEYA8 co-incubated with platelets for two hours ($n = 2$). **b** Top 10 down- and upregulated proteins (+plts vs. control) after RPPA analysis. **c** Heat map depicting differentially regulated genes in HEYA8 or HEYA8 cells co-incubated with $100 \times 10^6$ platelets under low-attachment conditions for 24 hours ($n = 3$, $p < 0.001$). **d** Enriched pathways with indicated $p$-values of upregulated genes in platelet co-incubated HEYA8 cells using Gene Set Enrichment Analysis (GSEA, www.broadinstitute.org/gsea). **e** Upstream analysis of gene expression changes in HEYA8 ovarian cancer cells co-incubated with platelets using Ingenuity Pathway Analysis (IPA, http://www.ingenuity.com/products/ipa). **f** Random walk analysis highlighting YAP1 protein as the main connector to transcriptional changes in HEYA8 co-incubated with platelets. **g** QRT–PCR analysis of YAP1, NDRG1, DDIT4, CDC20, CTGF, MCM5, MCM6, and MCM10 in HEYA8 cells after 24 and 48 h of co-incubation with platelets. 18S was used as the housekeeping gene ($n = 3$, two-sided Student's $t$-test). **h** QRT–PCR analysis of YAP1, NDRG1, DDIT4, CDC20, CTGF, MCM5, MCM6, and MCM10 after YAP1 knockdown with two different siRNAs (72 hours after transfection, $n = 3$). 18S was used as the housekeeping gene ($n = 3$, two-sided Student's $t$-test). *Bars* and *error bars* represent mean values and the corresponding SEMs (*$p < 0.05$, **$p < 0.01$, ***$p < 0.001$)

inducing anoikis resistance and in metastatic spread of cancer cells intraperitoneally and hematogenously by inducing a *YAP1*-dependent transcriptional program in detached cancer cells, which promotes cell survival and metastasis. Furthermore, the results suggest that reducing blood platelet counts or interfering with YAP1 signaling might be an important approach to limit ovarian cancer metastasis.

## Results

**Platelet-induced metastasis and anoikis protection.** To examine whether platelet numbers can impact survival of human ovarian cancer cells in the ascites, MDAH-2774 human ovarian cancer cells were injected intraperitoneally into nude mice. Prior to injection of cancer cells, one group of mice received an anti-platelet antibody (APA) to reduce platelet numbers before cell injection, whereas the second group of mice received a control antibody (ctrl IgG). Efficiency of APA at a dose of $0.5 \, \mu g \, g^{-1}$ in reducing platelet counts, and specificity of APA for platelets compared to other cell types were demonstrated previously[3, 4]. Treatment with ctrl IgG or APA was repeated twice weekly for the duration of the experiment. Reducing the number of platelets significantly increased the number of apoptotic cells in ascites, as measured by a higher number of SYTOX Red positivity in the APA as compared to that in the control group (Fig. 1a). To determine whether similar findings would apply to other cancer models, we used the human colon cancer cell line SW620 as a second model system. Here, reduced platelet counts also significantly diminished the number of nodules detected in the liver of mice after intrasplenic injection of SW620 cells (Fig. 1b, c and Supplementary Fig. 1a). No significant bleeding events occurred and mouse weight did not change after reduction in platelet counts (Supplementary Fig. 1b). These results highlight platelets as important contributors to metastasis in ovarian and colorectal cancer in vivo models.

To investigate whether platelets can have a direct impact on anoikis rates of ovarian and colon cancer cells in vitro, we co-incubated various human cancer cell lines with platelets under anoikis conditions using ultra-low attachment cell culture plates. After determining the baseline anoikis rate in all of the tested cell lines, we decided to use 72 h as the major endpoint for measuring anoikis levels in vitro (Supplementary Fig. 1c). Increase in the number of co-incubated platelets gradually reduced anoikis rates in HEYA8 human ovarian cancer cells (Fig. 1d). Similar observations were made for the effect of platelets on the anoikis rates of OVCAR8, SKOV3, MDAH-2774 and SW620 cells incubated with $100 \times 10^6$ platelets (Fig. 1e), as well as of A2780ip2 and OVCAR4 cells (Supplementary Fig. 1d) in which platelets significantly increased the number of surviving detached cells after 72 h. Moreover, platelet co-incubation with HEYA8 and OVCAR8 cells markedly reduced the percentage of cells in the sub-G1 phase of the cell cycle (Fig. 1f) and reduced protein levels of cleaved PARP (Fig. 1g). Additionally, western blot analysis of HEYA8 ovarian cancer cells co-incubated with platelets for 72 h under low-attachment conditions revealed a significant reduction in cleaved caspase-3 (Supplementary Fig. 1e), confirming a reduction in apoptosis of HEYA8 cells after platelet exposure. In contrast, anoikis rates of OVCA432, OVCAR5 and RKO cells did not change after platelet co-incubation, indicating a cell-specific responsiveness (Supplementary Fig. 1f). In summary, these results indicate that platelets induce anoikis resistance in vitro and enhance metastatic spread of tumor cells in vivo.

**Platelet-induced *YAP1* gene signature in cancer cells.** Next, we sought to identify the dominant signaling events in cancer cells responsible for platelet-mediated anoikis resistance. First, we

performed reverse phase protein array (RPPA) analysis to identify proteins and signaling pathways that were significantly altered in tumor cells after addition of platelets (Fig. 2a and Supplementary Data 1). We identified several proteins related to survival and proliferation that were upregulated, including pAkt$^{S473}$, p38$^{T180\_Y182}$ and pSrc$^{Y416}$ (Fig. 2b, Supplementary Fig. 2a). Interestingly, the strongest difference was observed for YAP1, with more than three-fold downregulation in the phosphorylation level at the serine 127 (S127) residue, indicating an activation of YAP1 signaling after platelet incubation (Fig. 2b). *YAP1* is a transcriptional co-activator that translocates into the nucleus after S127 dephosphorylation. Hence, to uncover transcriptional changes, we performed unbiased RNA expression analyses and isolated RNA from tumor cells incubated for 24 h with buffer only, or with platelets (Fig. 2c). Gene Set Enrichment Analysis (GSEA) revealed that pathways related to cell cycle and *E2F1*-signaling were the most upregulated pathways (Fig. 2d and Supplementary Fig. 2b), whereas genes related to hypoxia, oxidative phosphorylation and p53 were downregulated in cancer cells upon platelet incubation (Supplementary Fig. 2c, d). Intriguingly, upstream analysis using Ingenuity Pathway Analysis (IPA) identified *YAP1* as the principal and most crucial upstream regulator of gene expression changes seen in HEYA8 cells after platelet co-incubation (Fig. 2e). Additional computational analyses using CCExplorer used the receptor and transcription factor lists as well as background network[11] to identify connections between differentially regulated proteins (from RPPA) and transcriptional changes (from microarray analysis). As shown in Fig. 2f, YAP1 was the only differentially regulated protein which significantly connected to differential RNA expression patterns. The red nodes represent signaling nodes that connect YAP1 with differently regulated receptors (marked in yellow), likely to be upstream of YAP1. The green nodes are other genes found by random walk analysis, representing either differentially expressed genes (DEGs) or the ones linking DEGs.

Interestingly, many of the top up- and downregulated genes are well-known *YAP1* target genes, including connective tissue growth factor; *CTGF*[12], DNA damage inducible transcript 4; *DDIT4*[13], cell-division cycle protein 20; *CDC20*[14] and the DNA replication proteins minichromosome maintenance complex components 5/6/10; *MCM5*, *MCM6* and *MCM10*[15]. We validated expression of these genes after co-incubation of HEYA8 cells with platelets after 24 and 48 h (Fig. 2g). Conversely, the same genes were significantly deregulated by knocking down *YAP1* with two individual siRNAs (Fig. 2h), indicating that platelets induce a *YAP1*-dependent gene signature in cancer cells.

It is known that *YAP1* and the Hippo signaling pathway are crucial for organ development[16, 17] and cancer in various organs such as the liver[18], pancreas[15, 19] and prostate[20, 21], however, the data about the role of the Hippo–YAP pathway in ovarian cancer biology and metastasis is recently emerging. Analyzing TCGA data from high-grade serous ovarian cancer (Supplementary Fig. 2e) indicated that expression of *YAP1* or components of the Hippo signaling pathways is mainly regulated by gene amplification and/or mRNA up/downregulation, whereas mutations in these genes are rare, which is consistent with other tumor types[22]. Moreover, survival analysis evaluating the same set of patient samples indicated that YAP1 protein expression significantly correlated with disease-free survival in ovarian cancer patients (Supplementary Fig. 2f). Finally, we collected baseline platelet counts from 358 stage III and IV high-grade serous ovarian cancer patients whose gene expression patterns were previously analyzed by The Cancer Genome Atlas Research Network[23] and applied a verified gene signature for *YAP1* activation, which was shown to significantly correlate with patient survival[24, 25]. When the patients were stratified using this algorithm, tumors

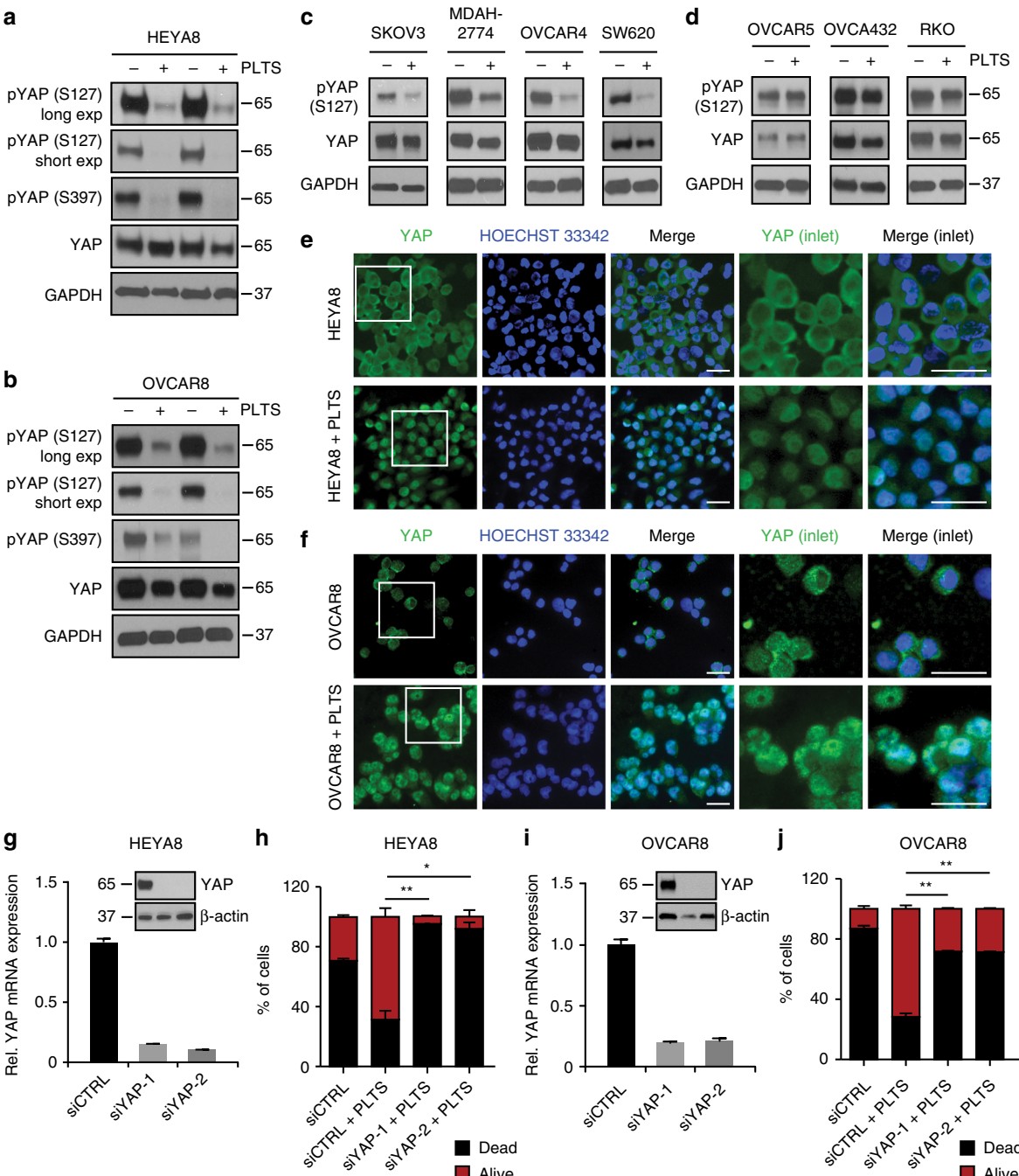

**Fig. 3** YAP1 is activated by platelets and is indispensable for platelet-induced anoikis resistance. **a**, **b** Western blot analysis of phosphorylated YAP1 (S127 and S397) and total YAP1 in HEYA8 **a** and OVCAR8 **b** cells after two hours under low-attachment conditions with or without platelet co-incubation. GAPDH was used as a loading control ($n = 5$). **c**, **d** Western blot analysis of phosphorylated YAP1 (S127) and total YAP1 in SKOV3, MDAH-2774, OVCAR4, SW620 **c** and OVCAR5, OVCA432 and RKO **d** GAPDH was used as a loading control ($n = 3$). **e**, **f** Immunofluorescence staining of YAP1 in HEYA8 **e** and OVCAR8 **f** cells after two hours under low-attachment conditions with (*lower panels*) or without (*upper panels*) platelet co-incubation. Inlets showing higher magnification of cells on the right side of the panels. Nuclear counterstain was done using Hoechst 33342 ($n = 3$). *Scale bars = 20 μm*. **g** QRT–PCR and western blot analysis in HEYA8 cells showing efficiency of YAP1 knockdown on the RNA and protein level using two different siRNAs ($n = 3$). **h** *Bar graphs* showing number of dead (SYTOX Red positive, *black*) and living (SYTOX Red negative, *red*) HEYA8 cells after 72 h of low attachment and 96 h after siRNA transfection ($n = 3$, two-sided Student's *t*-test). **i** QRT–PCR and western blot analysis in OVCAR8 cells showing efficiency of YAP1 knockdown on the RNA and protein level using two different siRNAs ($n = 3$). **j** *Bar graphs* showing number of dead (SYTOX Red positive, *black*) and living (SYTOX Red negative, *red*) OVCAR8 cells after 72 h of low attachment and 96 h after siRNA transfection ($n = 3$, two-sided Student's *t*-test). *Bars and error bars* represent mean values and the corresponding SEMs (\*$p < 0.05$, \*\*$p < 0.01$, \*\*\*$p < 0.001$)

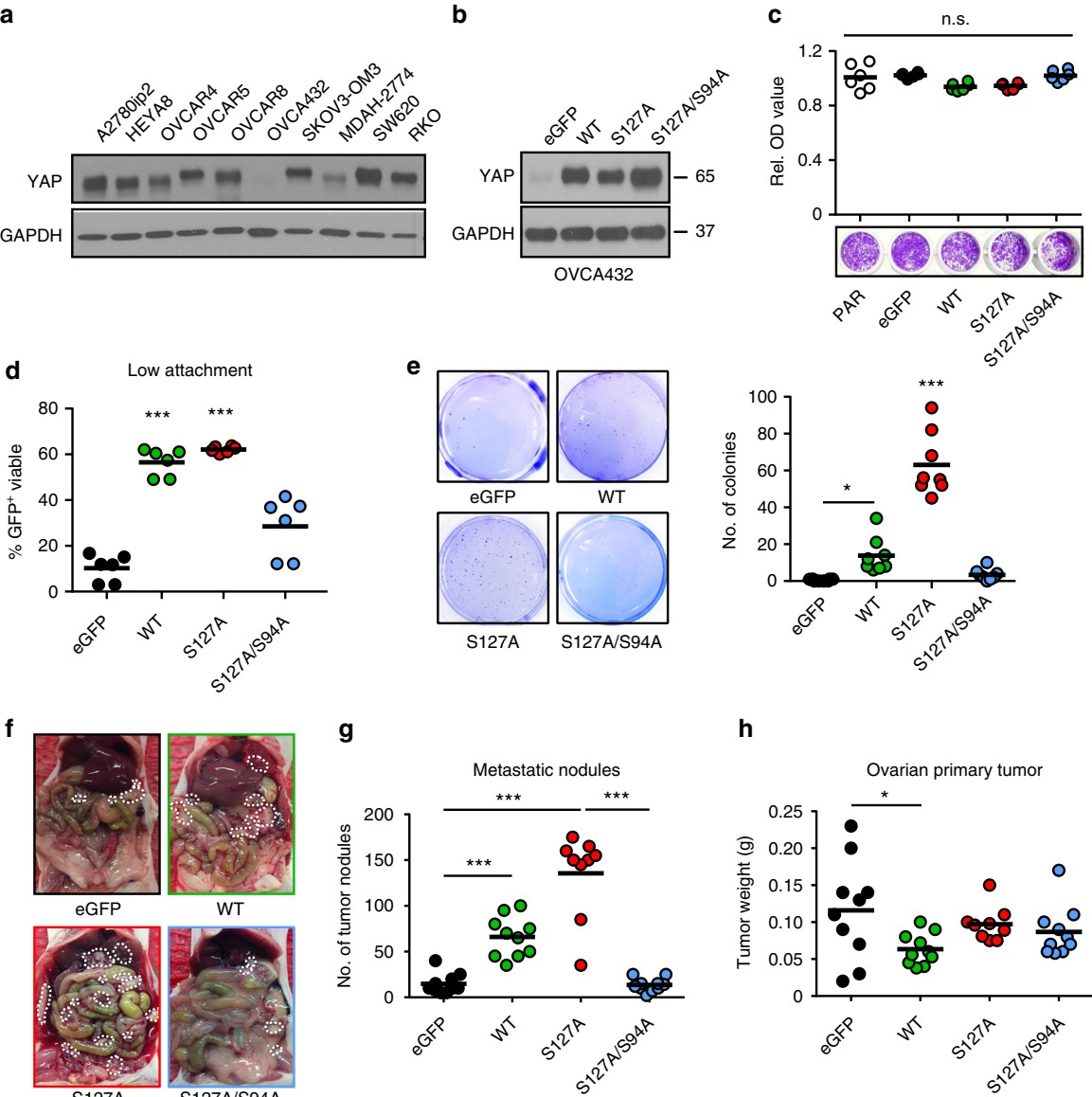

**Fig. 4** YAP1 overexpression increases anoikis resistance and metastasis. **a** Analysis of YAP1 protein expression in ovarian and colon cancer cell lines. GAPDH was used as a loading control ($n = 2$). **b** Validation of overexpression of YAP1 with wild-type (WT) YAP1, constitutively active (S127A) YAP1 and TEAD-defective mutant (S127A/S94A) YAP1 in OVCA432 human ovarian cancer cells. GAPDH was used as a loading control ($n = 3$). **c** Cell viability analysis (*upper panel*) and representative crystal violet staining (*lower panel*) of OVCA432 cells transduced with control or YAP1 overexpressing constructs 72 h after seeding into 96-well plates ($n = 6$, one-way ANOVA followed by a Tukey's multiple comparison post hoc test). **d** Analysis of GFP-positive, viable (SYTOX Red negative) cells after 72 h of low attachment in OVCA432 cells transduced with control or YAP1 overexpressing constructs ($n = 6$, one-way ANOVA followed by a Tukey's multiple comparison post hoc test). **e** Anchorage-independent growth assay in soft agar using OVCA432 transduced with control or YAP1 overexpressing constructs and quantification of colonies after seeding 15,000 cells and incubation for 14 days ($n = 3$, one-way ANOVA followed by a Tukey's multiple comparison post-hoc test). **f–h** Representative necropsy pictures **f**, number of metastatic nodules **g** and weight of ovarian primary tumor **h** after intraovarian injection of OVCA432 overexpressed with either eGFP-control, YAP$^{WT}$, YAP$^{S127A}$ or YAP$^{S127A/S94A}$ constructs ($n = 10$, one-way ANOVA followed by a Tukey's multiple comparison post-hoc test). *Bars* and *error bars* represent mean values and the corresponding SEMs (*$p < 0.05$, **$p < 0.01$, ***$p < 0.001$)

of 165 patients showed a *YAP1* activation signature. Interestingly, these patients had significantly higher platelet counts compared to patients whose tumors lacked this signature ($p = 0.04$, Supplementary Fig. 2g).

**YAP1 is required for platelet-induced anoikis resistance.** YAP1 activity is controlled by S127 and S397 phosphorylation. If hypophosphorylated, YAP1 translocates into the nucleus and binds to other transcription factors such as *E2F1* and *TEAD2/4*,

promoting transcription of genes that in turn increase proliferation and inhibit apoptosis[15, 26, 27]. Next, we analyzed the phosphorylation and intracellular localization of *YAP1*. Co-incubation of HEYA8 and OVCAR8 ovarian cancer cells with platelets robustly reduced YAP1$^{S127}$ and YAP1$^{S397}$ phosphorylation, whereas total YAP1 levels did not change (Fig. 3a, b). These results confirmed phosphorylation changes observed in the RPPA analysis. Reduced S127 phosphorylation was additionally validated in a panel of ovarian and colon cancer cell lines, including SKOV3, MDAH-2774, OVCAR4 and SW620 (Fig. 3c),

while OVCAR5, OVCA432 and RKO did not show this effect (Fig. 3d). Interestingly, the latter three cell lines were not protected from anoikis by platelet co-incubation (Supplementary Fig. 1f), implicating an important role of *YAP1* in platelet-mediated anoikis protection. Various ovarian cancer cell lines were not different in their ability to activate platelets in vitro (Supplementary Fig. 3a). Intriguingly, our experiment with OVCAR5 ovarian cancer cells confirmed that these cells are not responsive to platelet counts in vivo. We did not observe any reduction in primary ovarian tumor weight or number of

metastatic nodules induced by intraovarian injection of OVCAR5 cells into mice treated with APA as compared to those treated with ctrl IgG (Supplementary Fig. 3b). In line with the observed dephosphorylation, co-incubation of HEYA8 and OVCAR8 cells with platelets induced a clear shift of YAP1 protein expression from the cytoplasmic to the nuclear compartment of the cell (Fig. 3e, f and Supplementary Fig. 3c, d). These results together with the observed gene expression signature strongly support the idea that *YAP1* activation is responsible for platelet-induced anoikis resistance. To obtain direct evidence supporting this

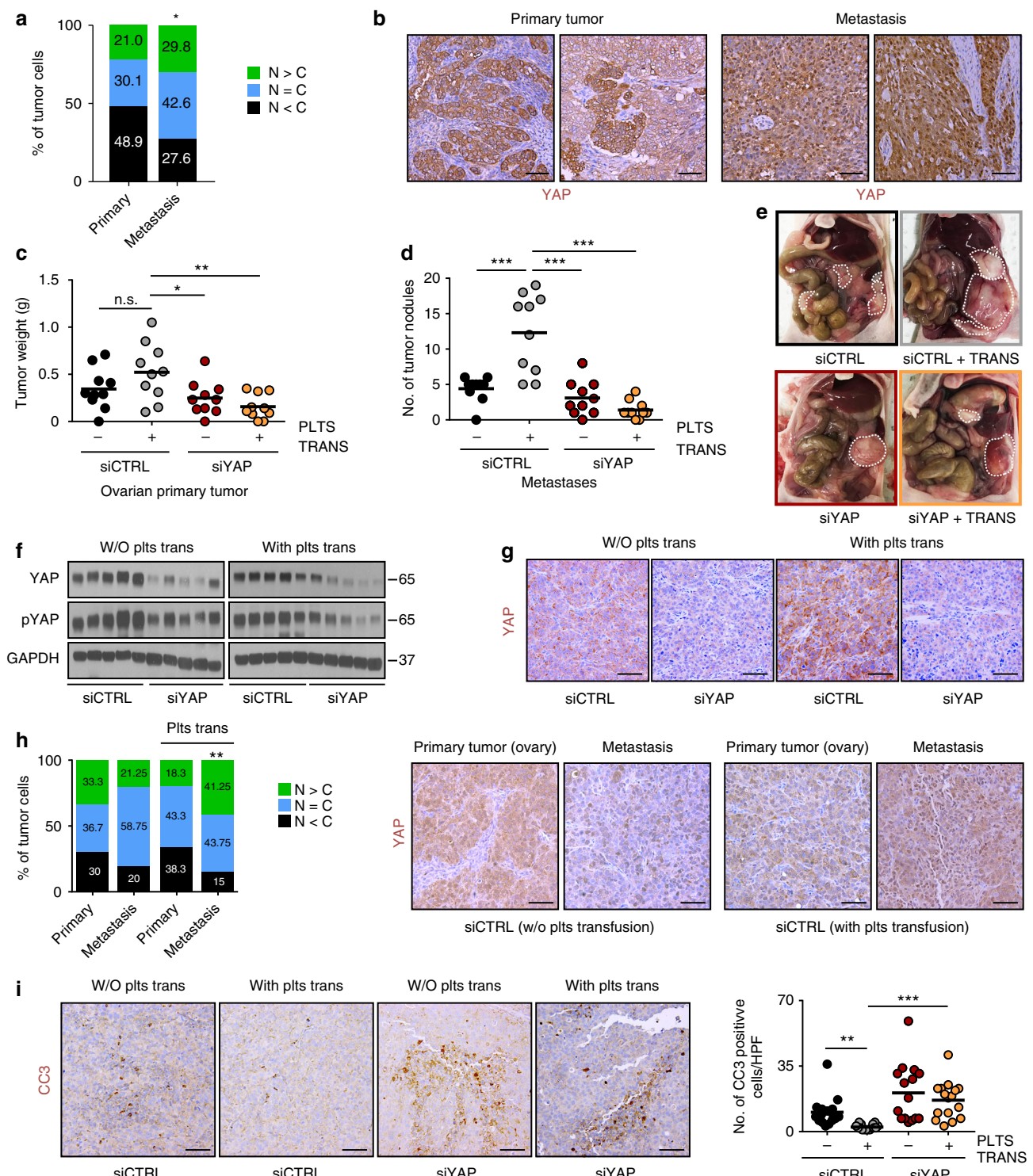

concept, we performed RNAi-mediated knockdown experiments in HEYA8 and OVCAR8 cancer cells using two independent siRNAs that reduced *YAP1* at RNA and protein levels (Fig. 3g, i). Twenty four hours after transfection of *YAP1* siRNA to ovarian cancer cells, platelets were added to the cells that were kept for an additional 72 h under low-attachment conditions. Analyzing the percentage of apoptotic cells via flow cytometry revealed that *YAP1*-depleted HEYA8 and OVCAR8 cells did not respond to platelets and showed the same apoptosis rate as the respective controls in which detached tumor cells were grown without platelets (Fig. 3h, j). Importantly, siRNA-mediated *YAP1* knockdown alone did not have a significant effect on anoikis rates in cancer cells (Supplementary Fig. 3e, f). Replication of these experiments in the SW620 colon cancer cells generated similar results (Supplementary Fig. 3g, h), suggesting that *YAP1* is an essential mediator of platelet-induced anoikis resistance in various types of cancer cells.

**YAP1 overexpression boosts anoikis resistance and metastasis.** Next, we assessed the effects of *YAP1* overexpression on anchorage-independent cell growth. Initially, we quantified YAP1 protein levels in various ovarian and colorectal cancer cell lines. This revealed a very low to absent protein expression of YAP1 in OVCA432 ovarian cancer cell line in comparison to a panel of different ovarian and colorectal cancer cells (Fig. 4a). Interestingly, this cell line exhibited very high anoikis rates at baseline levels with more than 50% apoptotic cells after only 24 hours and did not respond to the addition of platelets (Supplementary Fig. 1c, f). We investigated whether overexpression of *YAP1* would be sufficient to enhance survival of OVCA432 cells grown under low-attachment conditions. We stably introduced wild-type YAP1 (WT), a constitutively active YAP1$^{S127A}$ mutant or a TEAD-binding defective YAP1 mutant$^{S127A/S94A}$ into OVCA432 cells, and confirmed their expression (Fig. 4b). Over-expression of *YAP1* with various constructs did not affect normal 2D cell growth in 96-well plates (Fig. 4c). In contrast, when cells were grown under low-attachment conditions, overexpression of YAP1$^{WT}$ and the YAP1$^{S127A}$ mutant significantly enhanced cell survival, whereas overexpression of the TEAD-binding defective YAP1$^{S127A/S94A}$ double mutant compromised this effect (Fig. 4d). Similar results were obtained in a soft agar colony formation assay that confirmed a higher anchorage-independent growth in YAP1$^{WT}$, and especially in YAP1$^{S127A}$ overexpressing cells (Fig. 4e). Moreover, co-incubation of YAP1$^{S127A}$ overexpressing OVCA432 cells under low-attachment conditions with platelets for 72 h did not change anoikis rates (Supplementary Fig. 4a), suggesting that YAP1 mediates anoikis resistance even in cells were YAP1 is normally absent. To confirm our in vitro findings,

we injected $1 \times 10^6$ OVCA432 overexpressed with either control, YAP1$^{WT}$, YAP1$^{S127A}$ or YAP1$^{S127A/S94A}$ vector plasmids into the left ovary of nude mice. Intriguingly, mice injected with either OVCA432-YAP1$^{WT}$ but especially with constitutive active OVA432-YAP1$^{S127A}$ harbored greatly increased number of metastatic nodules (Fig. 4f, g and Supplementary Fig. 4b) as well as amount of ascites (Supplementary Fig. 4c) compared to control OVCA432 or OVCA432-YAP1$^{S127A/S94A}$. Primary tumor weight was not significantly different between the different groups despite a slight increase in primary tumor weight in the control group compared to mice injected with OVCA432-YAP1$^{WT}$ (Fig. 4h). YAP1 phosphorylation, in part, is regulated by Hippo and LATS kinases[28]. To understand the role of the upstream Hippo signaling pathway in the platelet-induced anoikis resistance, we analyzed the expression levels of LATS1, pLATS1, LATS2, MST1, MST2 and MOB1. Co-incubation of platelets with HEYA8 and OVCAR8 cells did not change levels of these proteins significantly (Supplementary Fig. 4d). Moreover, siRNA-mediated knockdown of the major YAP1 kinases *LATS1* and *LATS2* (Supplementary Fig. 4e) had no effect on anoikis resistance of HEYA8 and OVCAR8 cells compared to control siRNA (Supplementary Fig. 4f). Altogether, these results support the importance of the role of *YAP1* in mediating anoikis resistance and anchorage-independent cell growth in vitro as well as metastasis in vivo, which seems to be independent of the upstream Hippo signaling pathway.

**Thrombocytosis-induced metastasis is regulated by YAP1.** Most patients with newly diagnosed ovarian cancer have wide-spread metastases, which represents a challenge for every-day clinical management[29, 30]. Increased nuclear YAP expression was found to be associated with poor patient prognosis[17]. However, the specific role of *YAP1* in ovarian cancer metastasis has not been evaluated so far. By analyzing protein expression and quantifying nuclear versus cytoplasmic levels of YAP1 protein in 21 matched primary high-grade serous ovarian carcinomas and metastatic nodules (Supplementary Table 1), we found that primary tumors showed cytoplasmic YAP1 protein expression in almost 50% of tumor cells. In contrast, more than 70% of tumor cells in metastatic tumor nodules showed primarily nuclear YAP1 protein expression (Fig. 5a, b). Thrombocytosis significantly correlates with increased metastasis[31, 32] and is a hallmark of many solid tumors including ovarian cancer. We have recently shown that transfused platelets infiltrate into tumor tissue and increase tumor weight after intraperitoneal injections of cancer cells[4]. To investigate the role of platelets and *YAP1* in ovarian cancer metastasis, we injected $2 \times 10^5$ HEYA8 human ovarian cancer cells into the left ovary of nude mice and generated

**Fig. 5** Inhibition of YAP1 in vivo impedes thrombocytosis-enhanced metastasis. **a**, **b** Immunohistochemical staining and quantification of nuclear versus cytoplasmic YAP1 protein expression in 21 matched primary high-grade serous ovarian carcinomas and metastatic nodules. N > C: nuclear YAP1 > cytoplasmic YAP1; N = C: nuclear YAP1 = cytoplasmic YAP1; N < C: nuclear YAP1 < cytoplasmic YAP1 (*n* = 21, two-sided Student's *t*-test). **c–e** Aggregate tumor weight of ovarian primary tumor **c**, number of metastatic tumor nodules **d** and representative necropsy pictures **e** in mice receiving control (siCTRL) or YAP1 (siYAP) siRNA with or without platelet transfusion (PLTS TRANS) twice weekly, respectively (*n* = 10, one-way ANOVA followed by a Tukey's multiple comparison post hoc test). **f**, **g** Validation of YAP1 knockdown using western blot analysis of whole tumor lysates **f** or immunohistochemical staining **g** with antibodies against YAP1 (for WB and IHC) and phosphorylated YAP1 (S127, for WB). GAPDH was used as a loading control (for WB). Representative immunohistochemical images and western blot after processing of tumors from 7 mice. **h** Immunohistochemical analysis and quantification of nuclear versus cytoplasmic YAP1 positive tumor cells in tumor sections of primary and metastatic nodules. N > C: nuclear YAP1 > cytoplasmic YAP1; N = C: nuclear YAP1 = cytoplasmic YAP1; N < C: nuclear YAP1 < cytoplasmic YAP1 (*n* = 7, two-sided Student's *t*-test). Representative immunohistochemical images after processing of primary and metastatic nodules from 7 mice. **i** Immunohistochemical staining for cleaved caspase 3 (CC3) and quantification of the number of CC3-positive cells per high power field (HPF) in tumors treated with control (siCTRL) or YAP1 (siYAP) siRNA with or without platelet transfusions twice weekly (*n* = 7, one-way ANOVA followed by a Tukey's multiple comparison post-hoc test). Representative immunohistochemical images after processing of tumors from seven mice. *Bars* and *error bars* represent mean values and the corresponding SEMs (**p* < 0.05, ***p* < 0.01, ****p* < 0.001). *Scale bars* = 50 μm

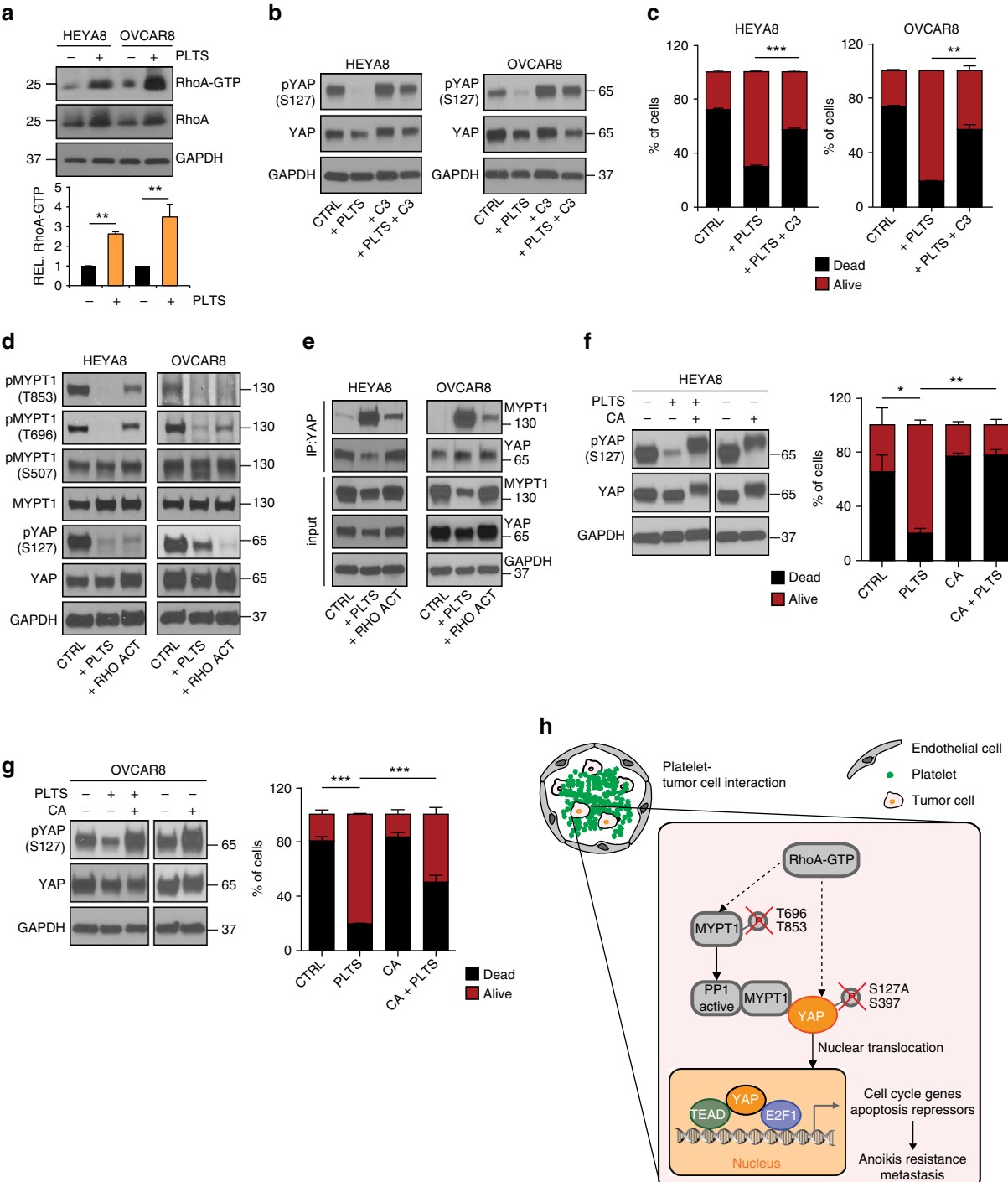

**Fig. 6** Regulation of YAP1 activity by RhoA-MYPT1-PP1 axis controls anoikis. **a** Analysis of active RhoA using RhoA-GTP pulldown assay in HEYA8 and OVCAR8 cells with or without platelet co-incubation for two hours. RhoA expression in input samples was used as control for quantification and GAPDH as loading control ($n = 2$, two-sided Student's $t$-test). **b** Protein analysis of phosphorylated (S127) and total YAP1 in HEYA8 and OVCAR8 cells after platelet co-incubation for two hours with or without 1.0 µg ml$^{-1}$ of Rho inhibitor C3 transferase ($n = 3$). **c** Bar graphs showing the number of dead (SYTOX Red positive, black) and living (SYTOX Red negative, red) HEYA8 or OVCAR8 ovarian cancer cells after growing for 72 hours in low attachment with or without platelet co-incubation and treatment with 1.0 µg ml$^{-1}$ of Rho inhibitor C3 transferase ($n = 3$, two-sided Student's $t$-test). **d** Protein analysis for phosphorylated MYPT1 (T853, T696 and S507), total MYPT1, phosphorylated YAP1 (S127) and total YAP1 in HEYA8 and OVCAR8 ovarian cancer cells after two hours of low attachment with platelet co-incubation or treatment of 1.0 U ml$^{-1}$ Rho activator ($n = 3$). **e** YAP1 co-immunoprecipitation assays showing MYPT1 interaction with YAP1 after platelet co-incubation or treatment with 1.0 U ml$^{-1}$ Rho activator for two hours under low-attachment conditions. Similar pulldown of YAP1 protein was confirmed. GAPDH was used as loading control ($n = 3$). **f, g** Protein analysis for phosphorylated (S127) and total YAP1 after platelet co-incubation for two hours with or without pre-treatment with 30 nM phosphatase inhibitor Calyculin A (CA) for 30 minutes in HEYA8 **f** and OVCAR8 **g** cells (left panels, $n = 3$). Bar graphs representing the number of dead (SYTOX Red positive, black) and living (SYTOX Red negative, red) HEYA8 **f** or OVCAR8 **g** ovarian cancer cells after platelet co-incubation for 72 h with or without pre-treatment with 30 nM phosphatase inhibitor Calyculin A (CA) for 30 min (right panels, $n = 3$, two-sided Student's $t$-test). **h** Proposed model of YAP1 activation, anoikis resistance and metastasis in cancer cells by platelets

thrombocytosis in mice by platelet transfusions twice weekly. Five days after tumor cell injections, treatments with the well-characterized 1,2-dioleoyl-sn-glycero-3-phosphatidylcholine (DOPC) nanoliposomes[2, 33, 34] carrying either non-targeting siRNA (siCTRL) or *YAP1* siRNA alone, or in conjunction with platelet transfusions were started and repeated twice weekly for 4 weeks. Intriguingly, our results indicate that platelet transfusion in siCTRL-DOPC-treated mice only slightly increased primary tumor growth in the ovary. This trend was, however, not significant (Fig. 5c). In stark contrast, the number of metastatic nodules (Fig. 5d, e and Supplementary Fig. 5a) and their aggregate tumor weight (Supplementary Fig. 5b) increased by up to 140% in platelet-transfused mice. Importantly, this effect of thrombocytosis on the number and total weight of metastatic nodules was completely abolished by simultaneous depletion of *YAP1*. Moreover, primary tumor weight, number of nodules and aggregate total tumor weight in the control and both *YAP1* siRNA-DOPC groups were comparable, irrespective of platelet transfusions (Fig. 5c, d and Supplementary Fig. 5b). Representative IVIS images are shown in Supplementary Fig. 5c. Mouse weights at the time of necropsy were not significantly different, suggesting that neither the injection of platelets nor *YAP1* siRNA-DOPC treatment had any obvious harmful effects in vivo (Supplementary Fig. 5d). To verify the efficiency of intraperitoneally injected *YAP1* siRNA in reducing *YAP1* in vivo, we performed protein analysis on resected tumors using Western blotting for total YAP1 and pYAP$^{S127}$ (Fig. 5f and Supplementary Fig. 5e, f) and immunohistochemistry for total YAP1 (Fig. 5g). These results confirmed efficient knockdown of *YAP1* in whole tumor lysates and histological tumor sections. Evaluation of nuclear versus cytoplasmic YAP1 in primary and metastatic tumor nodules receiving platelet transfusions or vehicle treatment revealed an up to 2.3-fold increase in the number of nuclear YAP1-positive cells in metastatic nodules in mice with thrombocytosis (Fig. 5h) compared to primary tumors and metastatic nodules from mice without thrombocytosis, further supporting the role of *YAP1* in metastatic spread of ovarian cancer. In addition, immunohistochemical analysis of tumors using antibodies against cleaved caspase 3 and Ki67 showed decreased apoptosis (Fig. 5i) and increased proliferation (Supplementary Fig. 5g) after platelet transfusion, whereas knockdown of *YAP1* in tumors reversed these effects.

**Platelet-induced RhoA increases MYPT1–YAP1 interaction**. Detachment of cancer cells from the extracellular matrix alters cellular architecture, focal adhesion formation and cytoskeletal arrangement[35]. The Rho family small GTPases play a key role in actin cytoskeleton organization, and RhoA has been shown to play a critical role in mediating the effect of cell attachment on YAP1 phosphorylation, likely through actin cytoskeleton organization[36]. To assess whether RhoA might also be involved in the platelet-induced dephosphorylation of *YAP1* in detached ovarian cancer cells, we performed RhoA pulldown activation assays in HEYA8 and OVCAR8 cells. Interestingly, platelets strongly increased RhoA activation in these tumor cells (Fig. 6a). Moreover, treatment of cells with the RhoA inhibitor botulinum toxin C3 (1.0 μg ml$^{-1}$) inhibited Rho activity (Supplementary Fig. 6a) and YAP1 dephosphorylation induced by platelet co-incubation while leaving total YAP1 unchanged (Fig. 6b). More importantly, subsequent analysis of anoikis levels in the same set of cells revealed that RhoA inhibition abolished the protective effect of platelets and increased anoikis levels compared to cells exposed to platelets alone (Fig. 6c). This effect was specific to the platelet-induced anoikis resistance since treatment of detached cells with the Rho inhibitor alone did not

change anoikis rates compared to untreated cells, both in HEYA8 and OVCAR8 cells (Supplementary Fig. 6b, c). In contrast, inhibition of Rho kinase (ROCK) with the small-molecule inhibitor Y-27632 did not prevent YAP1 dephosphorylation induced by platelet co-incubation in HEYA8 and OVCAR8 (Supplementary Fig. 6d) and had no influence on platelet-induced anoikis resistance (Supplementary Fig. 6e, f). These results suggest that platelets enhance RhoA activation in detached cancer cells, which in turn activates *YAP1* by promoting its dephosphorylation and leading to higher anoikis resistance.

PP1 and PP2 are two protein phosphatase family members that have been shown to dephosphorylate YAP1/2[37, 38]. The myosin phosphatase target subunit 1 (*MYPT1* or also called *PPP1R12A*) is a crucial regulatory subunit of myosin phosphatase PP1, and has been shown to interact with activated RhoA[39]. *MYPT1* is a well-known PP1-interacting protein, which enhances the specificity of PP1 for various substrates[40, 41]. MYPT1-PP1 activity is controlled through inhibitory phosphorylation of MYPT1 at threonine 696 (T696) and 853 (T853)[42]. Interestingly, MYPT1-PP1 was previously shown to regulate *NF2/Merlin* phosphorylation thereby modulating the Hippo signaling pathway[43, 44]. Hence, we hypothesized that the platelet-induced YAP1 dephosphorylation might be due to a RhoA-stimulated activation of the MYPT1-PP1 phosphatase leading to an enhanced interaction with YAP1. We investigated whether platelets alone or a direct activation of the RhoA signaling pathway would modulate *MYPT1* phosphorylation in detached HEYA8 and OVCAR8 cells. We either co-incubated the cancer cells with platelets or treated cancer cells with 1 U ml$^{-1}$ of calpeptin, an activator of Rho family GTPases[45]. Treatment of cells with calpeptin induced a robust RhoA activation (Supplementary Fig. 6g). After 2 h, protein lysates were collected and immunoblot analysis indicated that similarly to platelet co-incubation, activation of RhoA by calpeptin reduced YAP1$^{S127}$ and the inhibitory phosphorylation of MYPT1 at T696 and T853 while leaving the phosphorylation of serine 507 in MYPT1, used as a negative control, unchanged (Fig. 6d). This suggests that MYPT1-PP1 activity is increased by platelet incubation. Interestingly, immunoprecipitation of YAP1 in HEYA8 and OVCAR8 cells after platelet co-incubation or RhoA activation revealed a substantial increase in YAP1-MYPT1 interaction (Fig. 6e). These data suggest that platelets can induce the intracellular phosphatase activity causing dephosphorylation of YAP1, which in turn activates its downstream signaling to enhance anoikis resistance. Consequently, we predicted that blocking of phosphatase activity would abolish the platelet effects and restore anoikis rates. In fact, pre-treatment of HEYA8 and OVCAR8 cells with 30 nM of the serine and threonine phosphatase inhibitor Calyculin A (CA) for 30 min prior to incubation for two hours with $100 \times 10^6$ platelets in low-attachment plates kept YAP1 in a hyperphosphorylated state (Fig. 6f, g, left panels). Importantly, analysis of anoikis rates after 72 hours revealed that phosphatase inhibitors diminished the protective effect of platelets against anoikis in cancer cells (Fig. 6c, d, right panels). This further indicates that platelets induce an active dephosphorylation of YAP1 in cancer cells, which in turn activates YAP1 signaling allowing survival under low-attachment conditions.

In conclusion, we have identified platelets as major mediators for survival of detached cancer cell in vitro and in vivo, and provided evidence that YAP1 activation is critical for metastasis in ovarian and colorectal cancer models, at least partly due to its essential role in mediating anoikis resistance. Mechanistically, we identified a function for platelets in regulating MYPT1-PP1 activity, and we detected a platelet-induced YAP1-MYPT1 interaction leading to an enhanced dephosphorylation of YAP1 causing its nuclear trans-localization, and subsequent gene

expression changes leading to inhibition of apoptosis and increase in cell survival. Inhibition of YAP1 or phosphatase activity holds potential to reduce metastatic spread and could have substantial clinical implications (Fig. 6h).

## Discussion

The interaction between tumor cells and their microenvironment has been recognized as an important factor in tumor metastasis. Here we provide evidence that platelets activate cancer cell survival pathways, which induce anoikis resistance, and promote peritoneal and hematogenous metastasis. This is consistent with the close association between platelets and extravasating tumor cells[46] or detached cancer cells in the peritoneal cavity[3], and with the ability of platelets to increase proliferation of adherent tumor cells[47]. Cancer cells can activate platelets through secretion of factors such as ADP[4]. This, in turn, leads to degranulation and release of pro-angiogenic and pro-tumorigenic factors[5]. Pre-clinical evidence supports the notion that platelets not only increase tumor growth, but also facilitate metastasis, in part, by inducing a mesenchymal-like phenotype in cancer cells[48] or by guiding the formation of an early metastatic niche[49]. However, little is known about the transcriptional and post-transcriptional changes in detached cancer cells upon platelet co-incubation. Interestingly, a recent post hoc analysis of large, randomized cardiovascular prevention trials provided evidence for a reduced risk for cancer metastasis in patients taking aspirin at doses sufficient to inhibit platelet function[50, 51]. However, despite the evidence for a role of platelets in tumor metastasis, prospective clinical trials have been rare, due to concerns that anti-platelet drugs affect normal platelet function, leading to bleeding complications[32]. Here, we showed that platelets interact with cancer cells in a low-attachment environment and induced a YAP1-dependent transcriptional program which enhanced cancer cell survival and metastasis in vitro and in vivo. Gene enrichment analysis identified genes related to E2F1 and cell cycle. This is in concordance with a recent publication, showing that YAP1 positively regulates an E2F-dependent proliferation program in HCC[52]. Despite some correlative and in vitro studies suggesting that YAP1 expression is associated with poor patient prognosis in ovarian cancer[53, 54], careful evaluation of the role of YAP1 in ovarian cancer metastasis in vivo has not been done so far. Here, we report that the percentage of nuclear YAP1 expression is significantly higher in metastatic nodules of ovarian cancer patients compared to the matched primary tumor. Additionally, ovarian cancer patients with a tumor gene signature indicative of YAP activation had significantly higher platelet counts at time of diagnosis compared to patients who lacked this kind of signature, further supporting a link between platelet counts and YAP1. Moreover, we show that thrombocytosis-induced metastasis is completely blunted by YAP1 inhibition in vivo. These results were confirmed in colon cancer models. Lastly, overexpression of YAP1 in low-expressing OVCA432 human ovarian cancer cells confirmed the importance of YAP1 for metastasis in vitro as well as in vivo. Platelet interaction with cancer cells led to activation of RhoA and YAP1 dephosphorylation via the PP1-MYPT1 phosphatase which induced nuclear localization of the protein. Interestingly, a recently published proteogenomic characterization of ovarian cancer cells has identified RhoA-mediated signaling as important mediator of short survival[55], invasion and apoptosis resistance[56, 57]. Once in the nucleus, YAP1 likely binds to the TEAD family of transcription factors, as inhibition of this interaction diminished anoikis resistance induced by platelets. This is consistent with earlier reports showing that YAP1 downstream function depends highly on TEAD2 and TEAD4[12]. Intriguingly, cells that were not protected from anoikis

by platelets also did not show YAP1 dephosphorylation, indicating that YAP1 activity is a major contributor to the platelet-mediated enhancement of cancer cell survival under detached conditions. Serine-threonine phosphatase activity has been linked to several pathologies, including diabetes, cancer, cardiovascular disorders and Alzheimer's disease[58–61] as well as cellular processes including RNA splicing and DNA replication[62, 63]. Hence, targeting protein phosphatases to manipulate phosphorylation and activity states of YAP1 and other proteins within a cell could be a reasonable approach[64], as targeting YAP1 itself has remained a challenging task and needs further exploration of upstream regulators as well as downstream effectors[65]. In future, in depth analysis of the platelet proteome will help to identify whether there is one major mediator released from platelets rather than the plethora of cytokines and growth factors stored in platelets that activates YAP1 and will provide insights into potential new approaches to inhibit thrombocytosis-induced metastasis in ovarian cancer.

## Methods

**Reagents**. Rho inhibitor I (cat. no. CT04) and Rho activator I (cat. no. CN01) was purchased from Cytoskeleton, Inc. and used at a concentration of $1.0\,\mu g\,ml^{-1}$ and $1.0\,U\,ml^{-1}$, respectively. Calyculin A was purchased from Santa Cruz Biotechnology, Inc. (cat. no. sc-24000) and used at a concentration of 30 nM. ROCK inhibitor Y-27632 was purchased from Cell Signaling Technology (cat. no. 13624) and was used at a concentration of $10\,\mu M$. Analysis of RhoA activation levels was performed using the RhoA G-LISA activation assay according to the manufacturer's instructions (Cytoskeleton, cat. no. BK124).

**Cell lines and siRNA transfections**. Cell lines used in this study were obtained from the institutional Cell Line Core laboratory or ATCC, OVCA432 was a kind gift from Dr. Ronny Drapkin (Dana-Farber/Harvard Cancer Center). Cell lines were routinely tested for mycoplasma contamination using the MycoAlert kit (Lonza, Basel, Switzerland). Cell lines were authenticated using STR DNA profiling by the Characterized Cell Line Core Facility at MD Anderson. Human ovarian cancer cell lines HEYA8, OVCAR8, SKOV3, SKOV3-OM3, MDAH-2774, OVCAR4, OVCAR5, OVCA432 and A2780ip2 were cultured in RPMI-1640 (Sigma) supplemented with 10% fetal bovine serum (FBS), 100 U $ml^{-1}$ penicillin and 100 mg $ml^{-1}$ streptomycin. The human ovarian cancer cell line MDAH-2774 was maintained in MEM medium supplemented with 5% FBS, 1 × sodium pyruvate (100 mM stock, Invitrogen, Carlsbad, CA) and 1 × non-essential amino acid (Gibco, Carlsbad, CA). The human colon cancer cell lines SW620 and RKO were grown in DMEM or MEM medium, respectively, supplemented with 10% FBS, 100 U $ml^{-1}$ penicillin and 100 mg $ml^{-1}$ streptomycin. Cells were maintained at 37°C in a humidified incubator infused with 20% $O_2$ and 5% $CO_2$. All siRNA transfections were done using RNAiMAX (Invitrogen, Carlsbad, CA) reagent using forward transfection protocol (40 nM f.c. siRNA) and media was changed after 6 hours. siRNAs for in vitro and in vivo use were purchased from Sigma-Aldrich (St Louis, MO, ctrl siRNA: GCGACAGCUGGGCUGAAUA[dT][dT], siYAP1-1: SASI_Hs01_00182403, siYAP1-2: SASI_Hs01_00182404, siLATS1: SASI_Hs01_00046128, siLATS2: SASI_Hs01_00158804). Luciferase-labeling of HEYA8 was performed using pGreenFire Lenti-Reporter plasmid (System Biosciences) and GFP-positive cells were sorted 72 h after transduction.

**Low attachment/anoikis assay in vitro**. On day 0, 500,000 cells were seeded in an ultra-low attachment 6-well plate (cat. no. 3471, Corning). Immediately thereafter, platelets were isolated from the blood of nude or C57BL/6 mice and $100 \times 10^6$ platelets were added to cancer cells. Two hours after platelet addition, cells were lysed for protein analysis. 72 h after platelet addition the number of SYTOX Red (cat. no. S34859, Life technologies) positive cells was evaluated by a BD FACS-CantoII. Alternatively, cells were fixed in 75% ethanol and subsequently analyzed for cell cycle after propidium iodide staining on a Gallios Cell Analyzer (Beckman Coulter Gallios). In indicated experiments, $1.0\,\mu g\,ml^{-1}$ of Rho inhibitor I, $1.0\,U\,ml^{-1}$ Rho activator or $10\,\mu M$ ROCK inhibitor Y-27632 was used and cells were collected either after 2–4 hours for protein analysis or 72 h for SYTOX Red analysis. For Calyculin A treatment, cells were pre-treated with 30 nM of drug and then seeded in low-attachment plates and collected after 2 hours for protein analysis or 72 hours for SYTOX Red analysis. For experiments involving siRNA transfections, indicated siRNAs were transfected (100 nM f.c.) using reverse transfection 24 h before collecting cells and seeding them in low-attachment plates. Thereafter, cells were co-incubated with platelets and collected after 72 h for SYTOX Red analysis.

**Mouse platelet isolation**. Platelets were isolated from mouse blood as previously described[4]. Briefly, whole blood was drawn from the inferior vena cava of

anaesthetized nude mice into a syringe pre-loaded with 1:9 v/v 3.8% sodium citrate and mixed 1:1 v/v with tyrodes buffer lacking $Mg^{2+}$ and $Ca^{2+}$. Blood was centrifuged at 1100 r.p.m. ($200 \times g$) for 6 min at room temperature. The platelet-rich plasma fraction was passed through a filtration column of Sepharose 2B beads (Sigma-Aldrich) loaded into a siliconized glass column with a 10 μm nylon net filter (Millipore, Billerica, MA). Cloudy eluents contained platelets were collected. Platelets were counted with a hemocytometer by phase-contrast microscopy at ×400 magnification and immediately used for subsequent experiments. For protein expression analysis, platelets were washed once with PBS and lysed in RIPA buffer containing phosphatase and protease inhibitors.

**FACS analysis of CD62P surface expression in platelets**. Platelets were isolated as described above, and approximately $5 \times 10^6$ platelets were added to $5 \times 10^3$ HEYA8 cells in a 96-well plate for 2 hours. Thereafter, cell-platelet mixture was collected, washed once with PBS, and resuspended in 50 μl of Tyrodes buffer. 5 μl of anti–GPIbβ Dylight 488 (cat. no. X488, Emfret) and 5 μl of anti-CD62P-PE (cat. no. M130-2, clone Wug.E9, Emfret) was added and incubated for 15 minutes in the dark. Thereafter, 400 μl of PBS was added to stop the staining reaction. FACS analysis was done on a Beckman Coulter Gallios Flow Cytometer using forward scatter gating (for particle size) and fluorescence gating ($GP1b\beta^+$ particles) to specifically analyze platelets.

**Microarray and computational analysis of microarray data**. RNA was isolated from indicated groups of cells using the mirVana RNA Isolation kit (Thermo Fisher Scientific) per the manufacturer's instructions. Microarrays and data analysis were done as previously described[66]. For GSEA and IPA, a list of significantly altered genes ($p < 0.001$) from the microarray analysis was uploaded and the GseaPreranked analysis using the 'Hallmarks' gene set and an upstream analysis was performed, respectively. For RPPA analysis protein $p$-values were obtained from $F$-test using linear model and those with $p < 0.05$ were selected as differentially regulated proteins (DRPs). Microarray gene expression data of control and co-cultured HEYA8 after 24 h were normalized and $t$-test was applied to get the $p$-values for each gene. False discovery rate (FDR) was then obtained by Benjamini-Hochberg procedure. We then used random walk method as described earlier[11] to predict the crosstalk signaling pathways between platelets to tumor cells. The transition probability matrix was defined as $p_{ij} = \frac{w_{ij}}{\sum_j w_{ij}}$, where

$w_{ij} = a_{ij} * q_j$. The first term $A = \{a_{ij}\}$ is the adjacent matrix of graph G obtained by integrating 220 pathways downloaded from KEGG and the second term $Q = \{q_j\}$ is the FDR of each gene obtained from microarray data. By starting from the DRPs with equal probability, we calculate the probability for each node at step $n$ as $g^n = g^{n-1} * P$ until $|g^n - g^{n-1}| < 1*e-5$. Top 50 genes with highest probability were selected and mapped. The major connected component was predicted as the signaling pathways regulated by platelet-tumor co-culture. Prediction of a YAP1 activation signature via analysis of gene expression patterns of ovarian cancer tumors was done as described[24, 25]. Gene expression data from MCF10A breast epithelial cells overexpressing human YAP1 were collected from two series of experiments (GSE10196 and GSE13218). Discovery of a YAP1-specific signature associated with prognosis of patients with ovarian cancer had been generated from two independent cohorts in previous studies[25]. In the previous study, gene expression data from the Peter MacCallum Cancer Center (PMC cohort, GSE9891) were used as discovery cohort and for refining the prognostic gene expression signature[67]. This gene expression signature was applied to the current study. Expression patterns of the 388 genes from the PMC cohort were combined with the expression patterns of the TCGA ovarian cancer patient cohort to form a classifier according to the compound covariate predictor (CCP) algorithm[68]. This algorithm estimates the probability that a particular sample belongs to the YAP1 subgroup. The miscalculation rate in this training set (PMC cohort) was estimated by leave-one-out cross-validation during training. We then directly applied the developed classifier to gene expression data from the TCGA ovarian cancer patient cohort. Platelet counts (at time of diagnosis) of 358 patients were collected and differences between tumors with a YAP1 activation signature versus tumors which lacked this signature was calculated using Student's t-test. For survival analysis the patients were grouped into percentiles according to RPPA levels. The Log-rank test was employed to determine the association between RPPA levels and disease-free survival and the Kaplan–Meyer method was used to generate survival curves. The cutoff point (YAP: 0.48) was recorded to significantly split (log-rank test, $p$-value < 0.05) the samples into low/high protein groups. Evaluation of mutations, gene amplifications, RNA/protein up- or downregulation of YAP1 and Hippo pathway components was done using cBioPortal (http://www.cbioportal.org/, Ovarian Serous Cystadenocarcinoma, TCGA, Provisional).

**In vivo experiments**. For all in vivo experiments, female athymic nude mice (NCr-nu) were used which were purchased from Taconic Farms (Hudson, NY). Mice were cared for according to guidelines set forth by the American Association for Accreditation of Laboratory Animal Care and the US Public Health Service policy on Human Care and Use of Laboratory Animals. All mouse studies were approved and supervised by The University of Texas MD Anderson Cancer Center Institutional Animal Care and Use Committee. All animals used were 8–12 weeks old at the time of cancer cell injection. For therapeutic experiments, 10 mice were

assigned to each treatment group, unless stated otherwise. This sample size gave 80% power to detect a 50% reduction in tumor weight with 95% confidence. Female nude mice were injected with indicated numbers of human ovarian or colorectal cancer cells into the left ovary, spleen or intraperitoneally. After intraovarian, intraperitoneal or intrasplenic cell injection, mice were randomly assigned to the different treatment groups. During the course of the experiment, the primary investigator was not blinded to the group allocation. At time of necropsy, all investigators were blinded to the group allocation to assess the outcome of the experiment. For all intraperitoneal, intrasplenic or intraovarian cell injections, cells were collected at 70–80% confluence using trypsin-EDTA, neutralized with FBS-containing media, washed and resuspended in the appropriate cell number in Hanks' balanced salt solution (HBSS; Gibco) before injection. $1 \times 10^6$ MDAH-2774 cells were injected intraperitoneally (200 μl per injection), $0.2 \times 10^5$ HEYA8 cells and $1 \times 10^6$ OVCA432 or OVCAR5 cells were injected into the ovary (50 μl per injection) and $2 \times 10^6$ SW620 cells were injected into the spleen (50 μl per injection). Mice were given sustained-release buprenorphine subcutaneously for pain management, and the clips were removed after one week when the incision was completely healed. SiRNA treatment and platelet transfusion was started 5 days after cell injection and was done twice weekly. For platelet transfusions, $600 \times 10^6$ platelets were injected per mouse, which were previously isolated from female athymic nude mice as described above. To deplete platelets in mice, animals received a rat anti-mouse monoclonal antibody directed against mouse GPIbα (#R300; Emfret Analytics, Eibelstadt, Germany) at a concentration of $0.5 \, \mu g \, g^{-1}$ twice weekly that causes irreversible Fc-independent platelet depletion within 60 min of administration without inducing platelet activation. A polyclonal non-immune rat IgG antibody was used as control (#C301; Emfret Analytics, Eibelstadt, Germany). IVIS bioluminescence imaging was done once a week in order to monitor tumor growth in vivo. In all experiments, once mice in any group became moribund, they were killed, and their tumors were collected. Tumor weight of primary tumor and number and localization of tumor nodules were recorded. Tumor tissue was either fixed in formalin for paraffin slides, frozen in optimal cutting temperature (OCT) media to prepare frozen slides, or snap-frozen for lysate preparation. For analysis of SYTOX Red positivity of EPCAM+ tumor cells ascites was harvested and tumor cells were stained using FITC-labeled antibody directed against CD326 (EPCAM, 1:11, cat. no. 130080301, Miltenyi Biotec) and Sytox Red was added.

**Liposome nanoparticle preparation**. Liposomal nanoparticles were designed as previously described[33]. SiRNAs for in vivo experiments were incorporated into the 1,2-dioleoyl-sn-glycero-3-phosphatidylcholine (DOPC) neutral nanoliposomes. DOPC and siRNA were mixed in the presence of excess tertiary butanol at a ratio of 1:10 (w/w) siRNA/DOPC. Tween-20 was added to the mixture in a ratio of 1:19 Tween-20:siRNA/DOPC. The mixture was vortexed, frozen in an acetone/dry ice bath, lyophilized and stored at −20°C until further use. Before in vivo administration, preparation was hydrated with PBS at room temperature at a concentration of 150 μg kg⁻¹ body weight per injection. 100 μl of siRNA solution was injected intraperitoneally twice weekly.

**Immunoblot analysis**. Lysates from cultured cells were prepared using modified RIPA buffer (50 mM Tris–HCl (pH 7.4), 150 mM NaCl, 1% Triton, 0.5% deoxycholate) plus phosphatase and proteins inhibitors. The protein concentrations were determined using a BCA Protein Assay Reagent kit (Pierce Biotechnology, Rockford, IL). 10–40 μg protein lysates were loaded and separated on SDS–PAGE. Proteins were transferred to a nitrocellulose membrane by electrophoresis (Bio-Rad Laboratories, Hercules, CA). Membranes were blocked with either 5% BSA (for phosphorylated proteins) or 5% anhydrous milk powder in tris-buffered saline (for total proteins) for 1 h and then incubated at 4°C overnight with respective primary antibodies: YAP (1 : 1000, cat. no. 14074, Cell Signaling), Phospho-YAP (1 : 1000, S127, cat. no. 13008, Cell Signaling), Phospho-YAP (1 : 1000, S397, cat. no. 13619, Cell Signaling), PARP (1 : 1000, cat. no. 9532, Cell Signaling), Cleaved Caspase-3 (Asp175, 1 : 1000, cat. no. 9661, Cell Signaling), LATS1 (1 : 1000, cat. no. 3477, Cell Signaling), Phospho-LATS1 (1 : 1000, S909, cat. no. 9157, Cell Signaling), LATS2 (1 : 1000, cat. no. 13646, Cell Signaling), MST1 (1 : 1000, cat. no. 3682, Cell Signaling), MST2 (1 : 500, cat. no. 3952, Cell Signaling), MOB1 (1:500, cat.no.13730, Cell Signaling), MYPT1 (1 : 1000, cat. no. 8574, Cell Signaling), Phospho-MYPT1 (1 : 1000, T853, cat. no. 4563, Cell Signaling), Phospho-MYPT1 (1 : 1000, T696, cat. no. 5763, Cell Signaling), Phospho-MYPT1 (1 : 1000, S507, cat. no. 3040, Cell Signaling), PKC alpha (1 : 2000, cat. no. ab32376, Abcam), Phospho-PKC alpha (1 : 1000, cat. no. ab23513, Abcam), Src (1 : 1000, cat. no. 2109, Cell Signaling), Phospho-Src (1 : 1000, cat. no. 2101, Cell Signaling), Akt (1 : 1000, cat. no. 9272, Cell Signaling), Phospho-Akt (1 : 1000, cat. no. 9271, Cell Signaling), p38-MAPK (1 : 1000, cat. no. 8690, Cell Signaling), Phospho-p38-MAPK (1 : 1000, cat. no. 4511, Cell Signaling). After washing with tris-buffered saline with Tween-20, the membranes were incubated with HRP-conjugated secondary antibody for 1 h at room temperature. Antibodies against β-actin (1 : 5000, cat. no. A5316, Sigma-Aldrich) or GAPDH (1 : 5000, cat. no. G8795, Sigma-Aldrich) were used as a loading control. Full blots are included in Supplementary Fig. 7.

**Reverse phase protein analysis**. Human protein lysates were isolated from cells and platelets as described above. Samples were diluted to $1 \mu g \mu l^{-1}$ in RIPA buffer, denatured by 1% SDS with β-mercaptoethanol and stored at −80 °C until further use. Reverse phase protein array (RPPA) was performed at the MD Anderson RPPA Core facility using 40 μg protein per sample. All antibodies were validated previously[69].

**Quantative reverse transcription –PCR analysis**. For mRNA quantification, total RNA was isolated using the Direct-zol RNA MiniPrep kit (Zymo Research Corp.) and cDNAs were synthesized using the Verso cDNA kit (Thermo Scientific) per the manufacturer's instructions. Analysis of mRNA levels was performed on a Mx3005P qPCR system (Agilent Technologies, CA). The primers used are described in Supplementary Table 2.

**Immunofluorescence and immunohistochemistry**. Cells were spun down on a coated glass coverslip (cat. no. 08-774-383, Fisher Scientific) and fixed in 4% paraformaldehyde for 20 min at room temperature. Cells were then washed with PBS twice and blocked in PBS/5% BSA/0.3% Triton X-100 for 60 min. Thereafter, primary anti-human YAP antibody (1 : 400, cat. no. 14074, Cell signaling) was added, diluted in PBS/1% BSA/0.3% Triton X-100 and left for incubation overnight at 4°C. Next day, 488-labeled goat anti-rabbit secondary antibody (1 : 1000, cat. no.) was added for two hours at room temperature. Hoechst 33342 (1 : 10,000, Molecular probes) was used for nuclear counterstain. Slides were mounted using propyl gallate (Fisher Scientific, Hampton, NH) and glass cover slips. For immunochemical evaluation of proliferation and apoptosis, anti-Ki67 antibody (1 : 200, cat. no. RB-90-43-P; Thermo Scientific) and anti-caspase 3 antibody (cat. no. CP229B; BioCare Medical) was used, as previously described[4]. Evaluation of YAP1 expression in the in vivo tumors was carried out using anti-human YAP antibody (1 : 400, cat. no. 14074, Cell signaling) according to the manufacturer's instructions. Signal was developed using the HRP substrate 3,3′-diaminobenzidine (DAB). Nuclear counterstain was done using hematoxylin. ki67 and caspase 3 positivity as well as number of nuclear YAP1-positive tumor cells was quantified in five randomly selected fields at ×200 magnification. Primary tumors and corresponding metastatic nodules of ten ovarian cancer patients were collected and used to analyze cytoplasmic and nuclear YAP1 protein expression. Percentage of cells having cytoplasmic versus nuclear YAP1 expression was evaluated in five randomly selected fields at ×200 magnification.

**YAP1 overexpression constructs**. YAP1 overexpression and control constructs were engineered as previously described[15] and kindly provided by Ronald DePinho (UT MD Anderson Cancer Center, Houston, TX). Virus production was performed by transfecting 293 T cells with either GFP controls or various YAP1 constructs cloned in the pHAGE lentivirus vector. 72 h after transfection of OVCA432, GFP-positive cells were selected by flow sorting and analyzed for YAP1 protein overexpression using Western blotting.

**MTT cell viability assay**. $2 \times 10^3$ parental OVCA432 or OVCA432 transduced with various YAP1 overexpression constructs were seeded in 96-well plates. 3 days later cell viability was measured using MTT cell proliferation assay and stained with Crystal Violet (0.2% in 80% methanol) overnight and de-stained with distilled water.

**Soft agar assay**. $15 \times 10^3$ OVCA432 were resuspended in regular RPMI-1640 medium containing 0.4% (w/v) low-melting agarose (cat. no. 18300012, Thermo Scientific) and plated on six-well plates containing solidified 1.2% agar. After agarose containing cells was solidified, 2 ml of regular RPMI-1640 medium was added per well. After 14 days of incubation, formed colonies were washed twice with PBS and fixed with acetone/methanol for 10 minutes at room temperature. After washing with PBS, colonies were stained with 0.005% crystal violet for 20 minutes and washed with water to reduce background staining. Colonies were photographed and counted using ImageJ.

**RhoA pulldown activation assay**. Activated GTP-bound RhoA in HEYA8 with or without co-incubation with platelets for two hours under low-attachment conditions was measured using the RhoA pulldown activation assay biochem kit (Cytoskeleton, Inc., cat. no. BK036), per manufacturer's recommendations using 250 μg protein lysates and 25 μg Rhotekin-RBD beads per reaction.

**YAP1 pulldown assay**. Cells co-incubated with platelets for two hours or treated with $1.0 \text{ U ml}^{-1}$ Rho activator were lysed in $1 \times$ NP-40 cell lysis buffer containing protease inhibitors. 200 μg protein lysate was used per reaction. Cell lysates were pre-cleared using 20 μl of protein A/G plus-agarose beads (cat. no. sc-2003, Santa Cruz Biotechnologies, Inc.) for 30 min with rotation at 4°C. Beads were spun down afterwards and the supernatant with the cell lysate was transferred to a fresh tubes. Primary anti-human YAP antibody (cat. no. 14074, Cell signaling) was added at a concentration of 1:50 and the reaction was incubated with rotation overnight at 4°C. On the next day, 20 μl beads were added and incubated for 2 h with rotation at

4°C. Beads were centrifuged and washed five times with 500 μl of $1 \times$ NP-40 lysis buffer. Beads were resuspended in 15 μl of $2 \times$ SDS sample buffer and heated for 5 minutes at 95°C. Lysates were loaded on a 4–20% SDS–PAGE gel and proteins were transferred to a nitrocellulose membrane.

**Statistics**. Statistical analysis was done using Excel and GraphPad Prism 6. Differences between two groups were evaluated using two-tailed Student's $t$-test or one-way analysis of variance, adjusting for multiple comparisons. Results are presented as the mean ± SEM. For all statistical analyses, $p < 0.05$ was considered statistically significant.

**Study approval**. Approvals for all experiments were obtained from the University of Texas at M.D. Anderson Cancer Center Institutional Review Board (IRB). All animal experiments were approved and supervised by the MDACC Institutional Animal Care and Use Committee.

**Data availability**. Gene expression data from MCF10A breast epithelial cells overexpressing human YAP1 were collected from two series of experiments (GSE10196 and GSE13218). Discovery of a YAP1-specific signature associated with prognosis of patients with ovarian cancer had been generated from two independent cohorts in previous studies[25]. In the previous study, gene expression data from the Peter MacCallum Cancer Center (PMC cohort, GSE9891) were used as discovery cohort and for refining the prognostic gene expression signature[67]. The microarray data have been deposited in the GEO database under accession code GSE95275 (https://www.ncbi.nlm.nih.gov/geo/query/acc.cgi?acc=GSE95275). All other remaining data are available within the Article and Supplementary Files, or available from the authors upon request.

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

## Acknowledgements

Portions of this work were supported by the National Institutes of Health (CA177909, CA016672, CA109298, UH2TR000943, P50 CA083639, and P50 CA098258), Cancer Prevention and Research Institute of Texas (RP110595, RP120214), Ovarian Cancer Research Fund, Inc. (Program Project Development Grant), The RGK Foundation, The Judi A Rees ovarian cancer research fund, Mr. and Mrs. Daniel P. Gordon, The Blanton-Davis Ovarian Cancer Research Program, American Cancer Society Research Professor Award, and the Frank McGraw Memorial Chair in Cancer Research (A.K.S. & V.A.K.). M.H. was supported by a fellowship of the Deutsche Forschungsgemeinschaft (DFG). J.-M.H. was supported by the NCI-DHHS-NIH T32 training grant (T32 CA101642). T.G was supported by the Odyssey Fellowship Program at the UT MD Anderson Cancer Center. R.R. was supported in part by the Russell and Diana Hawkins Family Foundation Discovery Fellowship. K.M.G. is supported by the Altman-Goldstein Discovery fellowship. We thank all members of the MD Anderson Flow Cytometry and Cellular Imaging Core Facility for excellent technical assistance. The core facility is funded by the National Cancer Institute (NCI) Cancer Center Support Grant P30CA16672.

## Author contributions

M.H. performed experiments, analyzed data and wrote the manuscript. M.L.T. performed experiments and analyzed data. T.G., S.P., J.M.H., Y.M.L., R.L.D., M.S.C., R.R., A.S.N., K.M.G., Y.W., and L.S.M. participated in in vitro and in vivo experiments. C.R-A. and G.L-B. designed and prepared nanoparticles for in vivo studies. J.S., S.T.W., S.Y.Y.,

J.-S.L., and C.I. performed computational analyses. W.H., B.Y.K., and D.A.L. provided patient data. J.L. provided patient samples. V.A.K. and A.K.S. supervised and designed the study and wrote the manuscript.

## Additional information

**Competing interests:** The authors declare no competing financial interests.

