## [Peer Review File · Nature Communications]

Reviewers' comments:

Reviewer #1 (Remarks to the Author):

In the manuscript by Haemmerle et al., the authors describe a novel and exciting new mechanism by which platelets inhibit anoikis in ovarian cancer cells and thereby function to promote metastasis. The authors present evidence that this anoikis inhibition is driven by a RhoA-PP1 mediated dephosphorylation of YAP1 which subsequently translocates to the nucleus and promotes the expression of pro-survival genes. Overall, the studies are elegantly designed and have unveiled interesting and compelling data to back their proposed mechanisms. However, despite my enthusiasm for the manuscript, there are multiple issues that need to be addressed prior to publication. These are detailed below:

1. In Figure 1A and 1B, the authors utilize APA to reduce platelet numbers but do not include a control to prove the antibody is indeed reducing platelets. In addition, the authors show (Supplementary Figure 1D) that OVCA432, OVCAR5, and RKO cells do not alter anoikis following coincubation with platelets. What happens to anoikis and metastasis (as seen in Figures 1A-C) when one of these non-responsive cell lines is treated with APA? If platelets are not impacting anoikis in these cell lines, APA treatment would presumably not impact metastasis in vivo with these cells. An experiment along these lines would ease concerns regarding whether the APA effects seen in Figure 1A-C are in fact due to the platelet-mediated inhibition of anoikis.
2. There are multiple instances throughout the manuscript where important controls are absent. Examples include a control to prove efficacy of siRNA-mediated knockdown in 2H, a positive control for C3 treatment in 6B, a positive control for the Rho Activator in 6D/E.
3. Quantitation of the images in Figure 3E and 3F regarding the co-localization of YAP to the nucleus is important.
4. There are inconsistent data presented regarding YAP expression in OVCA432 cells. In Figure 3D, abundant YAP is present while no YAP is observed in Figure 4A. How do the authors reconcile these findings?
5. Why did the authors choose to utilize the OVCA432 cell line in Figure 4? While it is stated that this is due to the lack of YAP expression (which is complicated by the issue mentioned in point 4), the overexpression of YAP in this cell line and subsequent analysis of downstream signaling through LATS1 and MST1 is done entirely in the absence of platelets. I understand that this allows an assessment of the different phospho-mutants of YAP1, but why not look at platelet-cancer cell co-culture experiments (where YAP dephosphorylation is clearly observed) and then examine activation of the Hippo pathway? These types of experiments are completed in HeyA8 and OVCAR8 cells, but are curiously absent from the OVCA432 cell experiments.
6. Anoikis is often measured using a Sytox assay which does not specifically measure apoptotic death. Cleaved caspase-3 is measured in Figure 5, but is absent as a marker in the earlier figures. This type of analysis is important to compliment the Sytox assays if one truly wants to conclude that this is an apoptotic death...
7. In Supplementary Figure 2G, the authors show that tumors with a YAP1 activation signature also have higher platelet counts. However, the change (while statistically significant) is not of substantial magnitude. Can the authors comment on or cite any evidence that the change in platelet count observed in these patients would be enough to biologically enhance platelet/cancer cell interactions?
8. What happens in the platelet co-culture assays when the experiments are done in attached conditions? Do platelets still protect from cell death in attached cells? Even if attached cells are not

exposed to any pro-apoptotic stimulus, it is important to know the baseline levels of cell death to understand how specific the platelet-cancer cell effect is to detached cells.

Reviewer #2 (Remarks to the Author):

This manuscript presents a convincing body of work that demonstrates a clear pathophysiological phenotype, that of platelets promoting anoikis resistance, leading to enhanced metastatic tumour behaviour of cancer cells in vivo; and links it to its putative underlying mechanism, that of the activation of YAP1 by its dephosphorylation, induced by factors elaborated from platelets presumably through their degranulation. This mechanism is shown to be Rho A dependent. Targeting YAP-1 activation through Rho A inhibition mechanisms or through YAP-1 targeted depletion prevented this prometastatic phenotype.

I think this work is novel, convincing and will influence thinking in the field, in particular the consideration of clinical strategies and preconceptions about the importance of thrombocytopenia and platelet function in cancer medicine.

I do have some points to consider that may add to the security of the data:

1. It is unclear if APA cross reacts directly with cancer cells to produce this phenotype or whether the effect is specific to platelet depletion, authors need further justification or more experiments
2. I was surprised that there was no control for platelets only in these experiments, e.g. RPPA had no platelet only analysis. I am left wondering about the contribution of platelet proteome and transcriptome to the RPPA and transcript data
3. In Fig 1E it says 1×10^6 whereas in text it says 100×10^6 platelets - please clarify
4. cannot find text reference to figure 4G - maybe this is just an oversight/typo?
5. Survival data show clear PFS difference in YAP-1 expression - but a factor controlling tumour dissemination might be expected to affect overall survival rather than just duration of response to therapy - what are the OS data, are they not presented because of lack of significance? if non significant please say so and hypothesise why that might be
6. On page 12 relating to Fig 6D calpeptin activated RhoA "strongly reduced" the inhibitory phosphorylation of MYPT1 at T696 and T853. This is clearly an overstatement - needs to be toned down.

Reviewer #3 (Remarks to the Author):

Comments:

A. Summary of the key results:

In this manuscript, Haemmerle et al. provided convincing in vitro and in vivo evidence that RhoA-MYPT1-PP1-YAP1 is the major signaling pathway mediating platelet-induced reduced anoikis and increased tumor metastasis.

B. Originality and interest:

Although mounting evidence suggest that increased platelets in thrombocytosis is predictive of poor survival of various cancers and cancer metastasis, the molecular mechanism is largely unknown, This studies provided the first molecular mechanism underlying platelet-tumor cell interaction in regulating resistance to anoikis and apoptosis during ovarian and colon cancer metastasis. YAP1 is a central player of the Hippo signaling pathway that plays important roles in size control, tumorigenesis, metastasis, cancer stem cells renewal and differentiation. This study provided the first evidence that YAP1 also play critical role in mediating platelet-enhanced tumor metastasis. These studies will have significant impact on targeting of tumor metastasis by disrupting platelet-tumor interacting through modulation of YAP1 in the future.

C. D. Data & methodology & appropriate use of statistic and treatment of uncertainty

The experiments are well designed and have no major flaw.

E. Conclusion:

This is a very important finding in elucidating a novel YAP1 signaling in mediating platelets-induced reduced anoikis and metastasis. Therefore, I recommend it to be accepted without further revision.

F. Suggested improvement

See above comments

G. Reference: OK

H. Clarity and context: OK

Reviewer #4 (Remarks to the Author):

There is now a considerable body of evidence that platelets are critical for regulation of solid tumour cell cancer metastasis. While this is confirmed in the present study, the major focus of the manuscript is on the mechanism of how this occurs. Gene expression and proteomic analyses revealed that platelets activate YAP1 in cancer cells contributing to anoikis resistance and metastasis. This occurred via a signalling pathway that involved RhoA-MYPT1-PP1- mediated dephosphorylation of serine127 in YAP1 and its translocation to the nucleus. Targeting YAP1 using a nanoparticle-based siRNA delivery platform diminished metastasis in the setting of high platelet count in vivo. This is a well performed study and the experimental findings support the conclusion that YAP1 is an important potential therapeutic target in ovarian and colon cancer.

1. Is the number of platelets used to co-incubate with cancer cells to observe reduction of anoikis reflective of the number found infiltrated in solid tumour or present in ascites?
2. Is the level of reduction of cleaved PARP significant (Fig 1G)? The decrease is not clear, particularly with OVCAR8?
3. Can the authors speculate on why platelets had no effects on OVACA432, OVCAR5 and RKO cells? How are these cells differed to the responsive ones? Do these differences correlate with the capacity of different cancer cell lines to activate platelets?
4. Does platelet derived-RNA contribute to the heat map (Fig 2)? Rowley JW et al indicated the presence of YAP1 in the RNA transcriptome of mouse platelets. doi: <https://doi.org/10.1182/blood-2011-03-339705>
5. Does the RhoA activation in the detached cancer cells by platelets lead to formation of a cancer initiating stem cell population?
6. Without YAP1wt, can cancer cells still increase EMT markers in the presence of platelets? Slug, Snail etc?

I waive anonymity for this review, Michael Berndt

Reviewers' comments:

Reviewer #1 (Remarks to the Author):

In the manuscript by Haemmerle et al., the authors describe a novel and exciting new mechanism by which platelets inhibit anoikis in ovarian cancer cells and thereby function to promote metastasis. The authors present evidence that this anoikis inhibition is driven by a RhoA-PP1 mediated dephosphorylation of YAP1 which subsequently translocates to the nucleus and promotes the expression of pro-survival genes. Overall, the studies are elegantly designed and have unveiled interesting and compelling data to back their proposed mechanisms. However, despite my enthusiasm for the manuscript, there are multiple issues that need to be addressed prior to publication. These are detailed below:

1. In Figure 1A and 1B, the authors utilize APA to reduce platelet numbers but do not include a control to prove the antibody is indeed reducing platelets. In addition, the authors show (Supplementary Figure 1D) that OVCA432, OVCAR5, and RKO cells do not alter anoikis following coincubation with platelets. What happens to anoikis and metastasis (as seen in Figures 1A-C) when one of these non-responsive cell lines is treated with APA? If platelets are not impacting anoikis in these cell lines, APA treatment would presumably not impact metastasis *in vivo* with these cells. An experiment along these lines would ease concerns regarding whether the APA effects seen in Figure 1A-C are in fact due to the platelet-mediated inhibition of anoikis.

Response: We thank the reviewer for raising this important point. We performed an additional *in vivo* experiment, where we injected the human ovarian cancer cell line OVCAR5 into the left ovary of mice. One group of mice received the anti-platelet antibody twice weekly during the whole course of the experiment, whereas the other group received control IgG. Analysis of metastatic nodules and tumor weight of primary ovarian tumor at time of necropsy showed no significant difference between the two indicated groups, confirming the *in vitro* observed non-responsiveness of this cell line towards platelets. These results are now included in Supplementary Fig. 3b.

2. There are multiple instances throughout the manuscript where important controls are absent. Examples include a control to prove efficacy of siRNA-mediated knockdown in 2H, a positive control for C3 treatment in 6B, a positive control for the Rho Activator in 6D/E.

Response: For confirming YAP knockdown, we included YAP expression levels in control-transfected and siYAP-transfected HeyA8 cells (most left three bars in Fig. 2h), indicating a more than 80% reduction in YAP mRNA level. Additionally, we used RhoA G-LISA (Cytoskeleton) to confirm that Rho activity is inhibited by treatment with the Rho inhibitor C3 transferase and activated when the cells were treated with the Rho activator calpeptin. These results are now included in Supplementary Fig. 6a and Supplementary Fig. 6g, respectively.

3. Quantitation of the images in Figure 3E and 3F regarding the co-localization of YAP to the nucleus is important.

Response: As suggested, we have now performed quantification of nuclear vs. cytoplasmic YAP in HeyA8 and OVCAR8 cells with and without platelet co-incubation under low attachment conditions. The bar graphs are included in Supplementary Fig. 3d and 3e for HeyA8 and OVCAR8 cells, respectively.

4. There are inconsistent data presented regarding YAP expression in OVCA432 cells. In

Figure 3D, abundant YAP is present while no YAP is observed in Figure 4A. How do the authors reconcile these findings?

Response: We apologize for the unclear presentation of our data. In Figure 4a, different ovarian and colorectal cancer cell lines are included in one Western blot, also including cell lines with a rather higher YAP1 expression (e.g. HeyA8 or OVCAR8 cells). In contrast, in Figure 3d, cell lysate prepared from OVCA432 cells cultured under low attachment conditions with or without platelet co-incubation was loaded on a separate SDS-PAGE and exposure time was adjusted accordingly. This confirms that expression of YAP in OVCA432 is very low, but not completely absent.

5. Why did the authors choose to utilize the OVCA432 cell line in Figure 4? While it is stated that this is due to the lack of YAP expression (which is complicated by the issue mentioned in point 4), the overexpression of YAP in this cell line and subsequent analysis of downstream signaling through LATS1 and MST1 is done entirely in the absence of platelets. I understand that this allows an assessment of the different phospho-mutants of YAP1, but why not look at platelet-cancer cell co-culture experiments (where YAP dephosphorylation is clearly observed) and then examine activation of the Hippo pathway? These types of experiments are completed in HeyA8 and OVCAR8 cells, but are curiously absent from the OVCA432 cell experiments.

Response: We chose to use OVCA432 cells for the overexpression experiments because of the low YAP expression in this cell line compared to other human ovarian cancer cell lines (as shown in Figure 4a, see explanation above). Additionally, we used different YAP mutants to show how YAP activation in human ovarian cancer cells influences metastasis *in vivo* and anoikis *in vitro*. Additionally, we performed an *in vitro* experiment, where we grew YAP-overexpressing OVCA432 cells (constitutively active YAP^{S127A}) under low attachment conditions and exposed them to platelets. As shown in Supplementary Fig. 1f, there is no difference in anoikis rates between OVCA432 with or without co-incubation with platelets. Overexpression of YAP1 in OVCA432 cells decreases the number of dead cells after 72 hours of low attachment, however, additional co-incubation with platelets did not change anoikis rates in these cells, indicating that YAP expression is an important driver of anoikis resistance in cells where YAP is normally absent. These results are now included in Supplementary Fig. 4a.

6. Anoikis is often measured using a Sytox assay which does not specifically measure apoptotic death. Cleaved caspase-3 is measured in Figure 5, but is absent as a marker in the earlier figures. This type of analysis is important to compliment the Sytox assays if one truly wants to conclude that this is an apoptotic death...

Response: We thank the reviewer for pointing out this important experiment. In addition to Sytox analyses, we have now performed Western Blot analysis of HeyA8 cells cultured under low attachment for 72 hours with or without platelets to show apoptotic death measured by the detection of cleaved caspase-3. These results are now included in Supplementary Fig. 1e.

7. In Supplementary Figure 2G, the authors show that tumors with a YAP1 activation signature also have higher platelet counts. However, the change (while statistically significant) is not of substantial magnitude. Can the authors comment on or cite any evidence that the change in platelet count observed in these patients would be enough to biologically enhance platelet/cancer cell interactions?

Response: In several clinical studies, platelet counts >400,000 in cancer patients was associated with a significantly worse prognosis compared to lower platelet counts. Interestingly, this association was not limited to ovarian cancer, and was observed in various types of cancer (cervical, gastric, renal, and even in glioblastoma multiforme). Lee et al. (Gynecol Oncol, 2011) identified preoperative platelet counts > 400,000 in 34.6% of 179 patients with ovarian cancer, and found that this modest increase in platelet counts was significant associated with a shorter survival. In another study on 291 patients with cervical cancer, pretreatment platelet counts > 400,000 was detected in 29.6% of patients. Patients without extrapelvic disease and platelet counts > 400,000 had a 55% greater chance of dying compared to those with lower platelet counts (Hernandez E et al., Gynecol Oncol, 2000). Platelet counts > 400,000 was detected in 25% of 700 patients with renal cancer (Suppiah R et al., Cancer, 2006), 19% of 153 patients with glioblastoma multiforme (Brockmann MA et al., Neuro Oncol, 2007), and 6.4% of 1593 patient with gastric cancer (Hwang SG et al., Eur J Surg Oncol, 2012). In all of these studies, platelet counts >400,000 were significantly associated with a shorter survival.

8. What happens in the platelet co-culture assays when the experiments are done in attached conditions? Do platelets still protect from cell death in attached cells? Even if attached cells are not exposed to any pro-apoptotic stimulus, it is important to know the baseline levels of cell death to understand how specific the platelet-cancer cell effect is to detached cells.

Response: Previously, we showed that platelets reduced apoptosis and increased proliferation rate in attached cancer cells (Cho, MS et al., Blood 2012, Bottsford-Miller J et al., Clin Cancer Res 2014, Haemmerle et al., J Clin Invest 2016). This is in accordance with the findings (by us and other groups) that platelets increased tumor growth and cancer cell proliferation *in vivo*. The baseline apoptotic rate of attached cancer cells was shown to be between 7.5 and 20% and was reduced by 1.5-2-fold following co-incubation with platelets. The findings in the current manuscript indicated that apoptosis rates of detached cancer cells were between 20 and 90% after 72 hours in low attachment conditions. Co-incubation with platelets decreased anoikis rates in cancer cells by 1.5 to up to 4-fold.

Reviewer #2 (Remarks to the Author):

This manuscript presents a convincing body of work that demonstrates a clear pathophysiological phenotype, that of platelets promoting anoikis resistance, leading to enhanced metastatic tumour behaviour of cancer cells in vivo; and links it to its putative underlying mechanism, that of the activation of YAP1 by its dephosphorylation, induced by factors elaborated from platelets presumably through their degranulation. This mechanism is shown to be Rho A dependent. Targeting YAP-1 activation through Rho A inhibition mechanisms or through YAP-1 targeted depletion prevented this prometastatic phenotype. I think this work is novel, convincing and will influence thinking in the field, in particular the consideration of clinical strategies and preconceptions about the importance of thrombocytopenia and platelet function in cancer medicine. I do have some points to consider that may add to the security of the data:

1. It is unclear if APA cross reacts directly with cancer cells to produce this phenotype or whether the effect is specific to platelet depletion, authors need further justification or more experiments

Response: Previously, we performed careful analyses of potential cross-reactivity of the used anti-platelet antibody with cancer cells, mouse endothelial cells and pericytes; and did not detect any binding of anti-platelet antibody to cancer cells, endothelial cells, or pericytes. These results are part of the supplementary material of a previously published manuscript (Stone RL et al., NEJM, 2012). Additionally, we analyzed platelet counts after treatment with different APA concentrations and decided to use a concentration of 0.5 µg/g for further in vivo experiments. With this concentration, we achieve a 50-60% reduction of platelets in the blood without getting a 'non-physiological' thrombocytopenia (Stone RL et al., NEJM 2012; Haemmerle et al., JCI 2016).

2. I was surprised that there was no control for platelets only in these experiments, e.g. RPPA had no platelet only analysis. I am left wondering about the contribution of platelet proteome and transcriptome to the RPPA and transcript data

Response: We thank the reviewer for raising this important point. We performed RPPA analyses of platelets alone as well. The data are now included in Supplementary Table 2. pYAP S127 has a very low expression level, suggesting that expression level in cells is truly downregulated in HeyA8 cells after two hours of platelet co-incubation. We did not perform RNA analysis of platelets alone, as we predicted that the RNA 'contamination' coming from platelets is very small due to the small amount of RNA which can be isolated from platelets (Hillman et al., J Thromb Haemost 2006). Additionally, we used human-specific microarrays from Illumina (HumanHT-12 v4 Expression BeadChip Kit) with which we expected minimal cross-reactivity with RNA of mouse platelets.

3. In Fig 1E it says 1 X 10⁶ whereas in text it says 100 X 10⁶ platelets - please clarify

Response: We thank the reviewer for pointing out this typing error, and we have corrected it accordingly.

4. cannot find text reference to figure 4G - maybe this is just an oversight/typo?

Response: We thank the reviewer for pointing out this mistake; we have now included the reference for Figure 4G in the text.

5. Survival data show clear PFS difference in YAP-1 expression - but a factor controlling tumour dissemination might be expected to affect overall survival rather than just duration of response to therapy - what are the OS data, are they not presented because of lack of significance? If non significant please say so and hypothesise why that might be.

Response: We thank the reviewer for this important point. We re-analyzed available data with regard to overall survival and YAP protein expression in patients with high-grade serous ovarian cancer, however, we did not observe any positive correlation. The reason for this might be the restrictive selection of patients whose tumors were finally analyzed within the TCGA consortium (e.g., all of these were patients who had debulking surgery upfront). Patients who underwent neoadjuvant therapy and also tend to have higher platelet counts and worse overall survival were not included in this analysis.

6. On page 12 relating to Fig 6D calpeptin activated RhoA "strongly reduced" the inhibitory phosphorylation of MYPT1 at T696 and T853. This is clearly an overstatement - needs to be toned down.

Reponse: We thank the reviewer for this comment. We rephrased the sentence.

Reviewer #3 (Remarks to the Author):

Comments:

A. Summary of the key results: In this manuscript, Haemmerle et al. provided convincing in vitro and in vivo evidence that RhoA-MYPT1-PP1-YAP1 is the major signaling pathway mediating platelet-induced reduced anoikis and increased tumor metastasis.

B. Originality and interest:

Although mounting evidence suggest that increased platelets in thrombocytosis is predictive of poor survival of various cancers and cancer metastasis, the molecular mechanism is largely unknown, This studies provided the first molecular mechanism underlying platelet-tumor cell interaction in regulating resistance to anoikis and apoptosis during ovarian and colon cancer metastasis. YAP1 is a central player of the Hippo signaling pathway that plays important roles in size control, tumorigenesis, metastasis, cancer stem cells renewal and differentiation. This study provided the first evidence that YAP1 also play critical role in mediating platelet-enhanced tumor metastasis. These studies will have significant impact on targeting of tumor metastasis by disrupting platelet-tumor interacting through modulation of YAP1 in the future.

C. D. Data & methodology & appropriate use of statistic and treatment of uncertainty

The experiments are well designed and have no major flaw.

E. Conclusion:

This is a very important finding in elucidating a novel YAP1 signaling in mediating platelets-induced reduced anoikis and metastasis. Therefore, I recommend it to be accepted without further revision.

F. Suggested improvement

See above comments

G. Reference: OK

H. Clarity and context: OK

Response: We are very appreciative for the positive comments of the reviewer.

Reviewer #4 (Remarks to the Author):

There is now a considerable body of evidence that platelets are critical for regulation of solid tumour cell cancer metastasis. While this is confirmed in the present study, the major focus of the manuscript is on the mechanism of how this occurs. Gene expression and proteomic analyses revealed that platelets activate YAP1 in cancer cells contributing to anoikis resistance and metastasis. This occurred via a signalling pathway that involved RhoA-MYPT1-PP1- mediated dephosphorylation of serine127 in YAP1 and its translocation to the nucleus. Targeting YAP1 using a nanoparticle-based siRNA delivery platform diminished metastasis in the setting of high platelet count *in vivo*. This is a well performed study and the experimental findings support the conclusion that YAP1 is an important potential therapeutic target in ovarian and colon cancer.

1. Is the number of platelets used to co-incubate with cancer cells to observe reduction of anoikis reflective of the number found infiltrated in solid tumour or present in ascites?
Response: Our own unpublished observations show that we can count approximately 3.4×10^5 platelets per ml of ascites. Moreover, a recent publication analyzing ascites samples of ovarian cancer patients indicated that on average 2×10^3 cancer cells/ml are present in ascites (Peterson et al., PNAS 2013). This results in a ratio of 170:1 (platelets:tumor cells). For our experiments we used a ratio of 200:1 (platelets:tumor cells), therefore, we believe that the observed reduction in anoikis is reflective of what we are likely to see *in vivo* in cancer patients.

2. Is the level of reduction of cleaved PARP significant (Fig 1G)? The decrease is not clear, particularly with OVCAR8?

Response: We thank the reviewer for this important point and we have now included quantification of cleaved and total PARP levels as a bar graph in Figure 1G.

3. Can the authors speculate on why platelets had no effects on OVACA432, OVCAR5 and RKO cells? How are these cells differed to the responsive ones? Do these differences correlate with the capacity of different cancer cell lines to activate platelets?

Response: We evaluated platelet activation by measuring expression of P-selectin (CD62P) on platelets incubated with OVCA432, OVCAR5, HeyA8 or OVCAR8 ovarian cancer cells, using flow cytometry. Consistent with our previous study (Haemmerle et al., JCI 2016), we observed a clear activation of platelets after co-incubation with cancer cells, however, we did not detect a significant difference in the percent of activated platelets between different cancer cell lines. Therefore, we hypothesized that the enhanced anoikis resistance in cancer cells after platelet co-incubation is determined by the extent of downregulation of YAP phosphorylation within cancer cells and not by the extent of cancer cell-induced platelet activation. These results are now included in Supplementary Fig. 3a.

4. Does platelet derived-RNA contribute to the heat map (Fig 2)? Rowley JW et al indicated the presence of YAP1 in the RNA transcriptome of mouse platelets. doi:<https://doi.org/10.1182/blood-2011-03-339705>

Response: We performed protein analysis on platelets alone; these results are now included in Supplementary Table 2. We detected very small amount of YAP and pYAP expression in platelets. We did not perform RNA analysis of platelets alone, as we expect that the RNA 'contamination' originated from platelets is very small (Hillman et al., J Thromb Haemost 2006). Moreover, we used human-specific microarrays from Illumina (HumanHT-12 v4

Expression BeadChip Kit) and expected minimal cross-reactivity with RNA of mouse platelets. Furthermore, we did not observe any difference in RNA expression in HeyA8 cells with or without platelets (see Figure 2G), indicating that the RNA was mainly derived from cancer cells.

5. Does the RhoA activation in the detached cancer cells by platelets lead to formation of a cancer initiating stem cell population?

Response: We thank the reviewer for raising this point. We performed Aldefluor assay in order to measure the number of ALDH1+ cells with platelet co-incubation and/or treatment with RhoA inhibitor and Rho activator. Co-incubation with platelets alone did not change the number of ALDH1+ cells; treatment with both the Rho activator and the Rho inhibitor increased the number of ALDH1+ HeyA8 and OVCAR8 cells, however, these numbers were not changed after co-incubation with platelets. Therefore, these findings suggest that Rho activation by platelets does not lead to the formation of a cancer initiating stem cell population *in vitro*. These results are shown below.

ALDEFLUOR ASSAY

6. Without YAP1wt, can cancer cells still increase EMT markers in the presence of platelets? Slug, Snail etc?

Response: We performed an additional experiment where we knocked down YAP in HeyA8 cells with two different siRNAs and co-incubated them with platelets for 48 hours under low attachment conditions. qRT-PCR analysis on EMT-related genes included E-Cadherin, Vimentin, TWIST1/2 and Zeb1/2 in the different groups are shown below. We observed downregulation of CDH1 but also EMT-inducing genes, like Twist1, Vimentin and Zeb1/2 after co-incubation with platelets. Additional knockdown of YAP did not significantly change expression of these genes. We realize that these results might be different *in vivo* and are in contrast to the previously published results by Labelle et al. (Cancer Cell, 2011), however, in contrast to the aforementioned study all of our cancer cell-platelet co-incubation studies were conducted under low attachment conditions and different sets of cell lines were used which might explain the discrepancy between the obtained results. We recognize that this is an interesting research question, which we will address in the near future.

qRT-PCR FOR EMT-RELATED GENES:
YAP KNOCKDOWN AND PLTS CO-INCUBATION

I waive anonymity for this review, Michael Berndt

REVIEWERS' COMMENTS:

Reviewer #1 (Remarks to the Author):

The authors have appropriately addressed all of my previous concerns. I now recommend the manuscript for publication.

Reviewer #2 (Remarks to the Author):

I am satisfied that the issues raised during review have now been addressed

Reviewer #4 (Remarks to the Author):

The issues I raised in my initial review have been adequately addressed. No additional comments.

Reviewers' comments:

Reviewer #1 (Remarks to the Author):

The authors have appropriately addressed all of my previous concerns. I now recommend the manuscript for publication.

Response: We are appreciative of the reviewer's comments.

Reviewer #2 (Remarks to the Author):

I am satisfied that the issues raised during review have now been addressed

Response: We are appreciative of the reviewer's comments.

Reviewer #4 (Remarks to the Author):

The issues I raised in my initial review have been adequately addressed. No additional comments.

Response: We are appreciative of the reviewer's comments.